# CryoGEN: Generative Energy-based Models for Cryogenic Electron Tomography Reconstruction

**Yunfei Teng**[†]   **Yuxuan Ren**[‡]   **Kai Chen**   **Xi Chen**   **Zhaoming Chen**   **Qiwei Ye**

Beijing Academy of Artificial Intelligence (BAAI)

{yfteng, yxren, kchen, xchen, zmchen, qwye}@baai.ac.cn

## ABSTRACT

Cryogenic electron tomography (Cryo-ET) is a powerful technique for visualizing subcellular structures in their native states. Nonetheless, its effectiveness is compromised by anisotropic resolution artifacts caused by the missing-wedge effect. To address this, IsoNet, a deep learning-based method, proposes iteratively reconstructing the missing-wedge information. While successful, IsoNet's dependence on recursive prediction updates often leads to training instability and model divergence. In this study, we introduce CryoGEN—an energy-based probabilistic model that not only mitigates resolution anisotropy but also removes the need for recursive subtomogram averaging, delivering an approximate $10\times$ speedup for training. Evaluations across various biological datasets, including immature HIV-1 virions and ribosomes, demonstrate that CryoGEN significantly enhances structural completeness and interpretability of the reconstructed samples.

## 1 INTRODUCTION

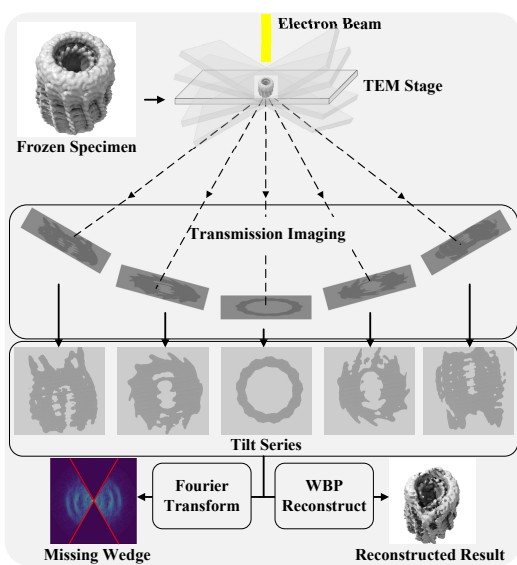

Figure 1: *Cryo-ET imaging and reconstruction.*

Cryo-ET is a cutting-edge imaging technique that allows the detailed examination of three-dimensional ($3D$) structures of biomolecules, cellular components, and whole organisms at near-atomic resolution, preserving them in a near-native and hydrated environment. Pioneering work by (Lucic et al., 2005) offers a comprehensive overview of using Cryo-ET to visualize cellular structures within their native environments. Subsequent advancements by (Beck & Baumeister, 2016) and (Schur et al., 2015) further demonstrate its capacity to resolve individual proteins and their interactions at subnanometer resolution. These capabilities position Cryo-ET as an indispensable tool for deciphering the intricate organization and regulation of molecular machinery in living systems.

The workflow of Cryo-ET proceeds as follows : a hydrated biological sample, intended for observation, is vitrified through rapid freezing to preserve its native state, and then continuously imaged by a transmission electron microscope (TEM) as the slide is incrementally tilted through a series of angles. This process will generate a series of two-dimensional projections, and the collection of these projections are commonly referred to as a *tilt series*. The tilt series are then aligned and reconstructed into a $3D$ structure, which is known as a *tomogram* in Cryo-ET. Among the various available tomographic reconstruction methods, weighted back-projection (WBP) (Radermacher, 2006) is the most widely used one. This entire procedure is depicted in Figure 1.

However, the optical constraints in the TEM stage typically limit the tilt range to approximately $\pm 60°$, resulting in a *missing wedge* of data in Fourier space. Consequently, tomograms reconstructed

---

†: Methodology development and algorithm implementation.
‡: Algorithm improvement and experimental validation.

via WBP exhibit anisotropic resolution (see Figure 2 (a)), with the lowest resolution observed along the $Z$-axis (parallel to the direction of electron beam). This anisotropy leads to distorted or elongated structural features in the $3D$ reconstruction, hindering the structural interpretation and underscoring a persistent challenge in Cryo-ET data analysis.

Traditional methods to mitigate the missing wedge problem include dual-axis tomography (Mastronarde, 1997), which acquires two perpendicular tilt series to reduce the missing wedge to a smaller "*missing pyramid*". Computational approaches such as model-based iterative reconstruction (Yan et al., 2019) and iterative compressed-sensing optimized nonuniform fast Fourier transform reconstruction (Deng et al., 2016) employ statistical tools and optimization techniques to compensate for missing data. However, these methods introduce additional complexity to either data acquisition or processing, yet they remain insufficient for fully restoring the missing information. Meanwhile, Cryo-ET contends with the inherent challenge of low signal-to-noise ratios (SNR) caused by the need to use low electron doses to prevent radiation damage. Traditional approaches, such as tailored filtering (Tegunov & Cramer, 2019) or contrast transfer function deconvolution (Cohen et al., 2024), can only partially mitigate noise, leaving necessary SNR enhancement as an unresolved challenge.

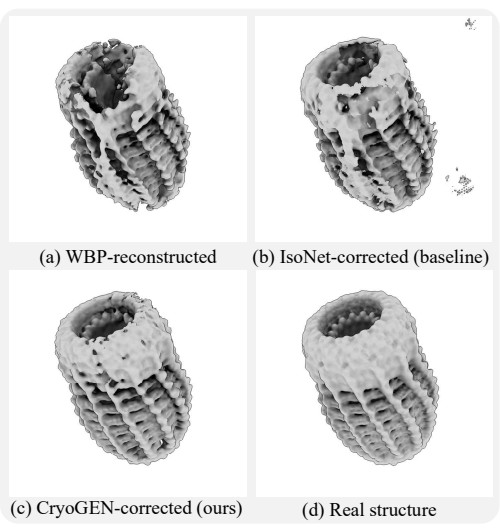

(a) WBP-reconstructed    (b) IsoNet-corrected (baseline)

(c) CryoGEN-corrected (ours)    (d) Real structure

Figure 2: *Reconstructed $3D$ structure comparison among (a) WBP (b) IsoNet and (c) our method using simulated Cryo-ET data of (d) C13 Vipp1 stacked rings (EMDB:18424).*

In the past few years, deep learning-based approaches have demonstrated promising avenues for restoring missing wedge data and improving SNR. IsoNet (Liu et al., 2022), for instance, has achieved notable success in restoring missing wedge information, inspiring subsequent methods (e.g., (Wiedemann & Heckel, 2024)). Concurrently, Noise2Noise-based denoising tools like CryoCARE (Buchholz et al., 2019), Topaz (Bepler et al., 2020), and Warp (Tegunov & Cramer, 2019), have significantly enhanced tomogram quality. Recent efforts (Wiedemann & Heckel, 2024; Zeng et al., 2024) aim to unify missing wedge correction and denoising into integrated pipelines. Despite these advances, current methods still produce artifacts and distortions, underscoring the need for more robust and refined solutions.

In this work, we advance Cryo-ET data reconstruction by introducing *CryoGEN*, a generative deep learning framework grounded in energy-based modeling. Different from IsoNet (Liu et al., 2022), which relies on recursive training susceptible to error accumulation and requiring iterative fine-tuning of predictions, CryoGEN eliminates these limitations through a direct modeling of data distributions from a Bayesian perspective. This approach leads to enhanced training stability, accelerated convergence, and immunity to pitfalls observed in IsoNet. Comparative results in Figure 2 (b) and (c) illustrate CryoGEN's superior reconstruction fidelity.

Our contributions are: *(1) Bayesian Formulation*: We frame Cryo-ET reconstruction within a Bayesian framework, addressing theoretical challenges and practical constraints; *(2) Objective Analysis*: We identify a potential failure mode in IsoNet's algorithm, analyze its underlying cause, and propose corrective strategies; *(3) Energy-Based Modeling*: We derive a novel objective function based on energy-based modeling and provide detailed end-to-end training and inference protocols for CryoGEN; *(4) Empirical Validation*: Across multiple datasets, we show that CryoGEN achieves significantly improved accuracy and faster training compared to existing methods.

## 2 PRELIMINARIES

### 2.1 PROBLEM FORMULATION

In Cryo-ET reconstruction, let $x \in \mathcal{X}$ represent the original data (complete tomograms with missing wedge restored), characterized by the distribution $p_x$. The observed data $y \in \mathcal{Y}$ corresponds to WBP-reconstructed tomograms affected by the missing wedge, following distribution $p_y$.

We also define a missing wedge operator $\mathcal{T}_M : \mathcal{X} \to \mathcal{Y}$, where $\mathcal{Y} = \{\mathcal{T}_M(x) \mid x \in \mathcal{X}\}$. This operator $\mathcal{T}_M$ acts as a many-to-one linear transformation, which can be conceptualized as first converting the image into the Fourier domain, masking to simulate tilt limitations, and transforming it back. The imaging process for Cryo-ET is modeled as:

$$y = \mathcal{T}_M(x) + \epsilon_n, \quad \epsilon_n \sim \mathcal{N}(0, \sigma_n^2 I), \tag{1}$$

where $\epsilon_n$ denotes additive Gaussian noise, with $\sigma_n^2$ determined experimentally. Consequently, Recovering $x$ from $y$ constitutes an ill-posed inverse problem.

### 2.2 BAYESIAN RULE

We observe that $y$ follows a Gaussian distribution $y \sim \mathcal{N}(\mathcal{T}_M(x), \sigma_n^2 I)$. Applying Bayes' theorem, we obtain $p(x|y) \propto p(y|x) \cdot p(x)$, which leads to:

$$\underbrace{\log p(x \mid y)}_{\text{Posterior}} = \underbrace{\log p(y \mid x)}_{\text{Likelihood}} + \underbrace{\log p(x)}_{\text{Prior}} + \underbrace{\text{constant}}_{\text{Evidence}}. \tag{2}$$

Thus, maximizing $\log p(x|y)$ is equivalent to maximizing the sum of $\log p(y|x)$ and $\log p(x)$. While the maximization of $\log p(y|x)$ is straightforward, the prior distribution $p(x)$ for complete tomograms is typically unknown. This gap not only presents a challenge but also necessitates the development of an effective method for properly defining this prior.

### 2.3 ENERGY-BASED MODELS

To model an arbitrary distribution, we employ Energy-based models (EBMs) (Lecun et al., 2006), which provide a probabilistic framework by defining distributions in terms of an *energy function*. We introduce a non-negative energy function $E_x$, allowing the distribution to be expressed as:

$$p(x) = \frac{1}{Z} \exp(-E_x(x)), \tag{3}$$

where $Z$ is the partition function (Boltzmann, 1974). The energy function $E_x$ assigns lower energies to high-probability states (e.g., properly reconstructed tomograms) and higher energies to implausible states, thereby sculpting the energy landscape to match the data manifold.

## 3 MOTIVATION

For each observation $y \in \mathcal{Y}$, substituting $\log p(y|x) = -\frac{1}{2\sigma_n^2}\|\mathcal{T}_M(x) - y\|_2^2$ into Equation (2) leads to the optimal reconstruction $x^*$ is obtained via maximum a posteriori (MAP) estimation as the following:

$$x^*(y) = \underset{x \in \mathcal{X}}{\arg\max} \log p(x|y) = \underset{x \in \mathcal{X}}{\arg\max} \left[ -\frac{1}{2\sigma_n^2}\|\mathcal{T}_M(x) - y\|_2^2 + \log p(x) \right]. \tag{4}$$

To circumvent the computational expense of iterative sampling or optimization, a natural strategy is to learn a parameterized mapping $g_\theta : y \mapsto x^*(y)$. This approach, which focuses on learning the mapping directly, is not immediately obvious but is fundamentally similar to the method (Johnson et al., 2016) (i.e., training neural networks to predict the targets directly rather than generating them iteratively during runtime). This concept also forms the foundation of IsoNet (Liu et al., 2022) and its variants (Buchholz et al., 2019), whose limitations we will discuss shortly, alongside the rationale for incorporating a generative model to address these challenges.

Consider a simplified scenario with a single observation $\mathcal{Y} = \{y_0\}$, where two distinct $x_1$ and $x_2$ map exactly to $y_0$ under the operation $\mathcal{T}_M$, so obviously:

$$\|\mathcal{T}_M(x_1) - y_0\|_2^2 = 0, \quad \|\mathcal{T}_M(x_2) - y_0\|_2^2 = 0.$$

Under the linearity of $\mathcal{T}_M$, the likelihood is maximized along the line segment connecting $x_1$ and $x_2$. Assuming a prior $p(x)$ modeled as a Gaussian mixture:

$$p(x) = \frac{1}{2} \left[ \mathcal{N}(x \mid x_1, \sigma^2 I) + \mathcal{N}(x \mid x_2, \sigma^2 I) \right],$$

where $\mathcal{N}(x|x_i, \sigma^2 I)$ denotes a single Gaussian distribution centered at $x_i$ with covariance matrix $\sigma^2 I$. Alternatively, this can be expressed as:

$$p(x) = \frac{1}{2\sqrt{2\pi\sigma^2}} \left[ \exp\left(-\frac{\|x - x_1\|^2}{2\sigma^2}\right) + \exp\left(-\frac{\|x - x_2\|^2}{2\sigma^2}\right) \right].$$

We assume $\sigma^2 < \frac{1}{5}\|x_1 - x_2\|_2^2$, ensuring sufficient separation between modes in $\mathcal{X}$, and seek to maximize the posterior $p(g_\theta(y_0)|y_0)$. Following IsoNet's methodology (as detailed in Appendix A.1), the objective simplifies to solving:

$$\theta' = \arg\min_\theta \frac{1}{2} \left( \|g_\theta(y_0) - x_1\|_2^2 + \|g_\theta(y_0) - x_2\|_2^2 \right).$$

This formulation seeks the parameter $\theta'$ that minimizes the mean squared error between the function's output $g_\theta(y_0)$ between both $x_1$ and $x_2$. Apparently, this leads to $g_{\theta'}(y_0) = \frac{1}{2}(x_1 + x_2)$, and thus

$$p\left(x = g_{\theta'}(y_0)\right) = \frac{1}{\sqrt{2\pi\sigma^2}} \exp\left(-\frac{\|x_1 - x_2\|^2}{8\sigma^2}\right). \tag{5}$$

This result may be unfavorable because

$$p\left(x = x_1 \text{ or } x = x_2\right) = \frac{1}{2\sqrt{2\pi\sigma^2}} \left[1 + \exp\left(-\frac{\|x_1 - x_2\|^2}{2\sigma^2}\right)\right]. \tag{6}$$

It can be shown that Equation (5) yields a lower probability compared to Equation (6) under our assumptions, indicating that the result learned by $g_{\theta'}(y_0)$ is suboptimal compared to directly selecting either $x_1$ or $x_2$. This typically happens when the probability density function of the prior distribution is *non-convex*, and thus simply averaging the minima lacks meaningful interpretation, highlighting a key limitation we have observed with IsoNet. By introducing an energy function as $E_x(x) = -\log p(x)$, the situation is depicted in Figure 3(a), where it is demonstrated that $E_x(\bar{x})$ does not correspond to a low-energy state, where $\bar{x}$ denotes the average of $x_1$ and $x_2$.

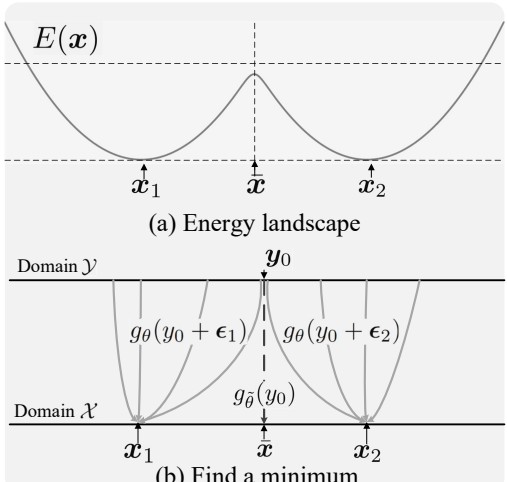

(a) Energy landscape

(b) Find a minimum

Figure 3: *Limitations of IsoNet formulation.*

The fundamental issue arises from the inherent one-to-many mapping, which precludes the existence of a deterministic function capable of associating a single observation with all potential minima. Mathematically, this reflects the constraint that the cardinality of $g_\theta(\mathcal{Y})$ cannot surpass that of $\mathcal{Y}$, inevitably causing undesirable results. A simple yet effective solution is to incorporate Gaussian noise through input augmentation by perturbing the input as $y + \epsilon_g$, where $\epsilon_g \sim \mathcal{N}(0, \sigma_g^2 I)$. Since the support of $\epsilon_g$ spans the entire real space (an uncountably infinite set), the ambiguity in the mapping is effectively resolved.

Integrating the energy model $E_x$ with $g_\theta(y + \epsilon_g)$ enables the generation of distributions aligned with the target, facilitating the identification of a suitable minimum rather than averaging over multiple minima. This method also circumvents the computational burden of one-the-fly sampling, inheriting the advantages of IsoNet while resolving the earlier limitation. As illustrated in Figure 3(b), the noise $\epsilon_g$ steers the model toward distinct low-energy states during the minimization of $E_x$, favoring convergence to a specific local minimum over an averaged solution. Notably, we later establish in Equation (9) that this framework can be reinterpreted as a generative task.

## 4 METHODOLOGY

Our objective is to determine a suitable energy model $E_x$, but training the energy model directly on the space $\mathcal{X}$ is not feasible due to the lack of labeled data, restricting training to the observation space $\mathcal{Y}$. However, we can assume that if a sample $x$ belongs to $\mathcal{X}$, its corresponding energy should be low. Moreover, we assume that if $x \in \mathcal{X}$, then $R(x) \in \mathcal{X}$ as well, where $R$ denotes a rotation operation selected from a predefined set of rotations $\mathcal{R}$ detailed in Appendix A.2. This rotation operation assumes that, aside from the missing wedge angles, the distributions are uniform in all directions, which is a fundamental premise in Cryo-ET reconstruction.

Thus we may train an energy model $E$ on $\mathcal{Y}$, with $E_x(x) = \frac{1}{|\mathcal{R}|} \sum_{R \in \mathcal{R}} E_y(\mathcal{T}_M(R \circ x))$. Inspired by (Lecun et al., 2006), we do not explicitly define the energy function. Instead, we could define it in a manner akin to contrastive learning, where low energy is assigned to samples while higher energy is attributed to other undesired regions in the energy space. This approach falls under the category of *implicit probabilistic models* (Diggle & Gratton, 1984). Using the trained energy model, we define the distribution $p_x$ based on the Boltzmann distribution, as previously described in Equation (3).

### 4.1 OBJECTIVE

We train the model with two objectives: *consistency loss* and *posterior maximization*. The consistency loss follows the IsoNet formulation; although it ultimately stabilizes, its inherent drawbacks can impede early progress and even cause divergence (see Section 3). The energy-based penalty, by contrast, supplies a sharper optimization signal but is well-known for instability at the end of training. To balance these trade-offs, we combine both terms.

#### 4.1.1 CONSISTENCY LOSS

In general, we can assume that $g_{\theta'}$ serves as an approximate inverse of $\mathcal{T}_M$. To enforce this inverse condition, we first introduce a consistency loss:

$$\textit{Consistency Loss} = \mathbb{E}_{y \sim p_y, \epsilon \sim \mathcal{N}(0, \sigma_h^2 I)} \left[ \frac{1}{|\mathcal{R}|} \sum_{R \in \mathcal{R}} \| R^{-1} \circ g_{\theta'} \circ \mathcal{T}_M \circ R \left( g_\theta(y) + \epsilon_h \right) - g_\theta(y) \|_2^2 \right],$$
(7)

where "$\circ$" denotes function composition. While Equation (7) largely mirrors IsoNet's objective function (Liu et al., 2022), it omits the epoch-wise recursive refinement step. If $\mathcal{T}_M$ is not one-to-one—as is often the case—the problem highlighted in Section 3 can arise, so injecting noise $\epsilon_h$ becomes essential.

#### 4.1.2 MAXIMUM A POSTERIOR

Next, our goal is to generate results that maximize the log posterior $\log p(x|y)$ by incorporating the previously discussed energy penalty term. The objective is to minimize the error on $\mathcal{X}$ using a model trained on the $\mathcal{Y}$. Therefore, we must ensure that $E(\mathcal{T}_M(x))$ is low when $\mathcal{T}_M(x) \in \mathcal{Y}$ and high otherwise. Therefore, we can define the posterior as:

$$\textit{Posterior} = \mathbb{E}_{y \sim p_y, \epsilon \sim \mathcal{N}(0, \sigma_g^2 I)} \left[ -\frac{1}{|\mathcal{R}|} \sum_{R \in \mathcal{R}} E \left( \mathcal{T}_M \circ R \circ g_\theta(y + \epsilon_g) \right) + \frac{1}{2\sigma_n^2} \| \mathcal{T}_M \circ g_\theta(y + \epsilon_g) - y \|_2^2 \right],$$
(8)

where $\sigma_n^2$ denotes the hyperparameter introduced in Equation (1). By overcoming the limitations of relying solely on the consistency loss, our approach obtains a notable performance gain in the early stage of training.

### 4.2 ENERGY MODEL

The energy model $E$ can be constructed using various methods, such as (Chen et al., 2020; Du et al., 2023). In this work, we directly employ generative adversarial networks (GANs) (Goodfellow et al., 2014). Specifically, we define our energy model as a parameterized neural network $E_\phi$ with parameters $\phi$. This configuration allows for the simultaneous training of both the energy model

$E_\phi$ and $g_\theta$. Consequently, the energy model is learned through adversarial training, as outlined in Equation (9):

$$Energy = \max_\phi \min_\theta \left[ \mathbb{E}_{y \sim p_y, \epsilon \sim p_\epsilon, \epsilon_g \sim \mathcal{N}(0, \sigma_g^2 I)} \frac{1}{|\mathcal{R}|} \sum_{R \in \mathcal{R}} E_\phi \left( \mathcal{T}_M \circ R \circ g_\theta(y + \epsilon_g) \right) - E_\phi(y + \epsilon) \right].$$
(9)

Furthermore, we consider a similar formulation as (Arjovsky & Bottou, 2017). Let $y$ follow the distribution $p_y$ with support on $\mathcal{Y}$, and let $\epsilon$ be an absolutely continuous distribution with density $p_\epsilon$. Then, the distribution $p_{y+\epsilon}$ is also absolutely continuous with density:

$$p_{y+\epsilon}(z) = \mathbb{E}_{y \sim p_y} \left[ p_\epsilon(z - y) \right] = \int_\mathcal{Y} p_\epsilon(z - y) \, dp_y.$$
(10)

Especially, If $\epsilon \sim \mathcal{N}(0, \sigma^2 I)$, then $p_{y+\epsilon}(z) \propto \int_\mathcal{Y} e^{-\frac{\|z - y\|^2}{2\sigma^2}} \, dp_y$, and Equation (9) reaches Nash equilibrium when $p_{g_\theta}(z) = p_{y+\epsilon}(z)$, as follows:

$$p_{g_\theta}(z) = \frac{1}{|\mathcal{R}|} \int_{y \in \mathcal{Y}} \int_{\epsilon_g \in \mathbb{R}^d} \sum_{R \in \mathcal{R}} \mathbb{1}_{z = \mathcal{T}_M \circ R \circ g_\theta(y + \epsilon_g)} \cdot p_{\epsilon_g}(\epsilon_g) \, p_y(y) \, d\epsilon_g \, dy.$$
(11)

This result is analogous to the conclusion presented in (Goodfellow et al., 2014), except that here, the data distribution is obtained by convolving the original data distribution with a Gaussian.

## 4.3 ALGORITHM

The CryoGEN algorithm consists of two stages: *training* and *inference*. (1) During the *training* stage, a prediction model $g_\theta$ is trained by a combination of consistency loss and posterior maximization along with an energy model $E_\phi$, using Equations (7) to (9). The full training procedure is detailed in algorithm 1; (2) In the *inference* stage, the complete tomogram is first divided into multiple overlapping subtomograms. These subtomograms are then processed through the prediction model $g_\theta$ to generate refined versions, which are later reassembled into a complete tomogram. Finally, to minimize edge effects, overlapping regions are averaged using a weighted approach.

---

**Algorithm 1** Training prediction model.

---

**Require:** Tomogram dataset $\mathcal{Y}$, noise levels $\sigma^2, \sigma_g^2, \sigma_h^2 > 0$, estimated noise variance $\sigma_n^2$, energy model $E_\phi : \mathbb{R}^d \to \mathbb{R}^+$, prediction model $g_\theta : \mathbb{R}^d \to \mathbb{R}^d$, learning rate $\eta$, energy penalty term $\lambda \geq 0$.

  **repeat**

    Randomly generate noise $\epsilon, \epsilon_g, \epsilon_h \sim \mathcal{N}(0, \sigma^2 I), \mathcal{N}(0, \sigma_g^2 I), \mathcal{N}(0, \sigma_h^2 I)$.

    Set $f(\phi, \theta) = -E_\phi \left( \mathcal{T}_M \circ R \circ g_\theta(y + \epsilon_g) \right) + 1/2\sigma_n^2 \cdot \| \mathcal{T}_M \circ g_\theta(y + \epsilon_g) - y \|_2^2$

    Update $\theta' \leftarrow \theta' - \eta \cdot \frac{\partial}{\partial \theta'} \| R^{-1} \circ g_{\theta'} \circ \mathcal{T}_M \circ R \left( g_\theta(y) + \epsilon_h \right) - [g_\theta(y)] \|_2^2$

    Update $\phi \leftarrow \phi - \eta\lambda \cdot \frac{\partial}{\partial \phi} [E_\phi(y + \epsilon) - f(\phi, \theta)]$

    Update $\theta \leftarrow \eta\lambda \cdot \theta + (1 - \eta\lambda) \cdot \theta' - \eta\lambda \cdot \frac{\partial}{\partial \theta} f(\phi, \theta)$

    Reduce the energy penalty term $\lambda$

  **until** convergence

  **return** $g_\theta$

---

## 5 EXPERIMENT

In this section, we compare our method with IsoNet, the state-of-the-art missing wedge correction technique, across various experiments. First, we validate our hypothesis using simple geometric shapes, as described in Section 5.1. Next, we assess our algorithm on a simulated dataset and benchmark it against other approaches in Section 5.2. Finally, we evaluate our method on real-world examples to test its robustness, as discussed in Section 5.3. Additional implementation and data processing details are provided in the Appendix, where we also present results from the latest simultaneous missing wedge correction and denoising method, DeepDeWedge (Wiedemann & Heckel, 2024).

*Remark.* CryoGEN achieves a significant speedup, completing the training process approximately $10\times$ faster than IsoNet. For instance, with an NVIDIA V100, CryoGEN requires only two hours, whereas IsoNet takes around 20 hours on Section 5.3 experiment.

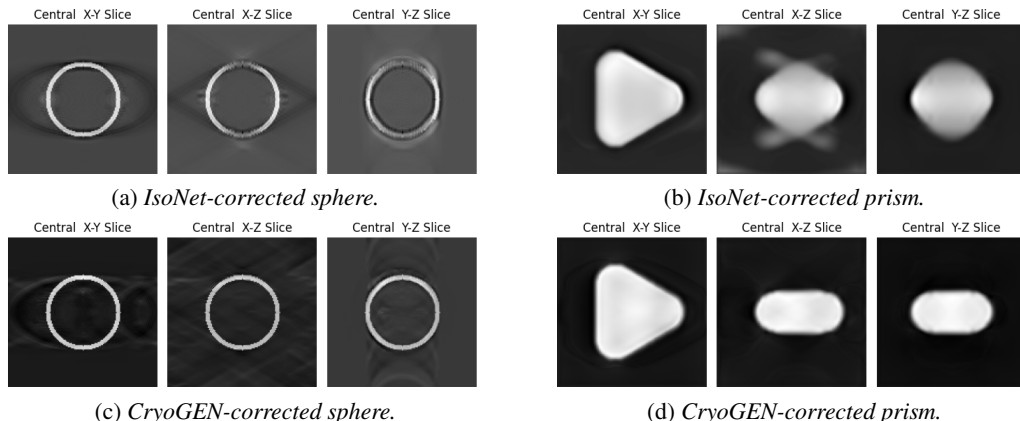

(a) *IsoNet-corrected sphere.*

(b) *IsoNet-corrected prism.*

(c) *CryoGEN-corrected sphere.*

(d) *CryoGEN-corrected prism.*

Figure 6: *CryoGEN and IsoNet corrected shapes.*

## 5.1 SIMPLE SHAPES

First, we validate the algorithm's effectiveness using simple synthetic shapes with known ground truth. We generate a $3D$ sphere and a triangular prism, artificially introducing a missing wedge. To illustrate this, we transform the $X$-$Z$ slice into the Fourier domain, clearly revealing the missing wedge, as shown in Figure 4.

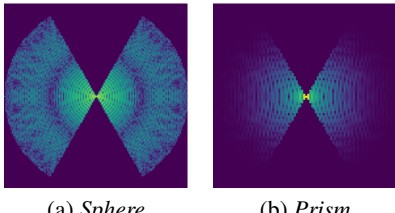

(a) *Sphere.*      (b) *Prism.*

Figure 4: *Central Fourier slice of the original missing-wedged shapes.*

Our objective is to reconstruct the missing wedge regions, and we demonstrate that our algorithm significantly outperforms the baseline in this task. As illustrated in Figure 5, the synthetic images exhibit reduced resolution in the directions affected by the missing wedge, while the $X$-$Y$ slice closely matches the ground truth as designed) closely matches the ground truth by design, as illustrated in Figure 5.

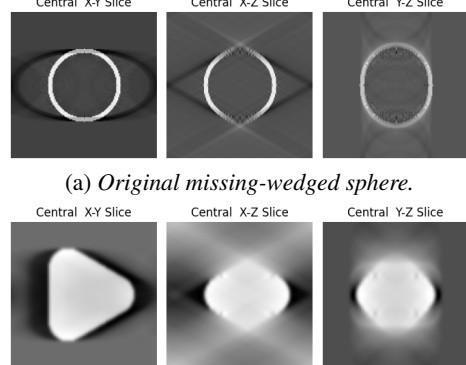

(a) *Original missing-wedged sphere.*

(b) *Original missing-wedged prism.*

Figure 5: *Original missing-wedged shapes.*

We first apply both IsoNet and CryoGEN to the two shapes, with the results shown in Figure 6. Both methods aim to restore the corrupted regions. While IsoNet successfully reconstructs the sphere, some artefacts remain, and it fails to restore the prism, producing a distorted oval in the $X$-$Z$ and $Y$-$Z$ slices. In contrast, CryoGEN achieves an almost perfect restoration of both shapes, with the slices closely resembling the original clean images. Our experiments reveal that CryoGEN can effectively recover more of the missing wedge regions and capture high-frequency signals.

Next, we quantitatively evaluate performance using the Peak Signal-to-Noise Ratio (PSNR) and Structural Similarity Index (SSIM) between the ground truth and the generated results. Definitions of PSNR and SSIM are provided in Appendix A.3.1. We compare the corrupted datasets, the results corrected by IsoNet and DeepDeWedge, and those corrected by CryoGEN. As shown in Table 1, our method consistently outperforms the baselines.

## 5.2 SIMULATED DATA

In this section, we applied our algorithm to more complex protein assemblies. Following the approach of IsoNet, we first evaluated our performance on the publicly available atomic model apoferritin (PDB:6Z6U) (Yip et al., 2020). Additionally, we selected the recently published electron microscopy dataset of C13 Vipp1 stacked rings (EMDB:18424) (Junglas et al., 2024). The results show that CryoGEN delivers more consistent outcomes in both the spatial and Fourier domains. Additionally, CryoGEN demands significantly less training time compared to IsoNet.

Table 1: *Quantitative evaluation of image quality for tomography reconstructions using different methods, comparing PSNR and SSIM metrics (**higher** values indicating better performance for both metrics) on sphere, prism and Vipp1 assembly datasets.*

| Data State | Sphere | | Prism | | Vipp1 assembly | |
|---|---|---|---|---|---|---|
| | PSNR | SSIM | PSNR | SSIM | PSNR | SSIM |
| Corrupted | 21.12 | 0.8113 | 14.82 | 0.6931 | 26.68 | 0.8000 |
| Iso-corrected | 22.98 | 0.8770 | 19.11 | 0.8857 | 27.12 | 0.8191 |
| Dewedge-corrected | 23.17 | 0.8824 | 21.10 | 0.9278 | 28.75 | 0.8758 |
| CryoGEN-corrected | **29.19** | **0.9706** | **32.69** | **0.9949** | **30.65** | **0.9199** |

**Apoferritin.** We made reconstructions using the atomic model of apoferritin (# PDB: 6Z6U), a widely-used benchmark in high-resolution CryoGEN. The simulated maps were then randomly rotated in ten different orientations, and a missing wedge was imposed in Fourier space, producing subtomograms with missing wedge artifacts. In this setup, CryoGEN achieved noticeably superior performance compared to IsoNet and also substantially reduced the training time. These improvements are particularly evident when inspecting the low-density volume in ChimeraX (Goddard et al., 2018), as illustrated in Figure 7.

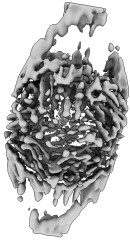 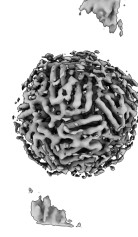 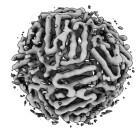

(a) *WBP reconstructed missing-wedged apoferritin, displaying both corrupted and missing regions in the low-density volume.*

(b) *Structure generated by IsoNet, displaying a corrupted region with visible inconsistencies in the low-density volume.*

(c) *Structure generated by Cryo-GEN, showing a much smoother and coherent low-density volume representation.*

Figure 7: *Comparison of low-density volumes generated by WBP (**a**), IsoNet (**b**) and CryoGEN (**c**). The results indicate that IsoNet produces synthetic artifacts stemming from the reconstruction of the missing wedge, which CryoGEN effectively resolves.*

**C13 Vipp1 Stacked Rings.** We assessed our method using the recently published C13 Vipp1 stacked rings dataset (# EMDB:18424), which features complex molecular assemblies. This dataset was obtained from the Electron Microscopy Data Bank (wwPDB Consortium, 2023).

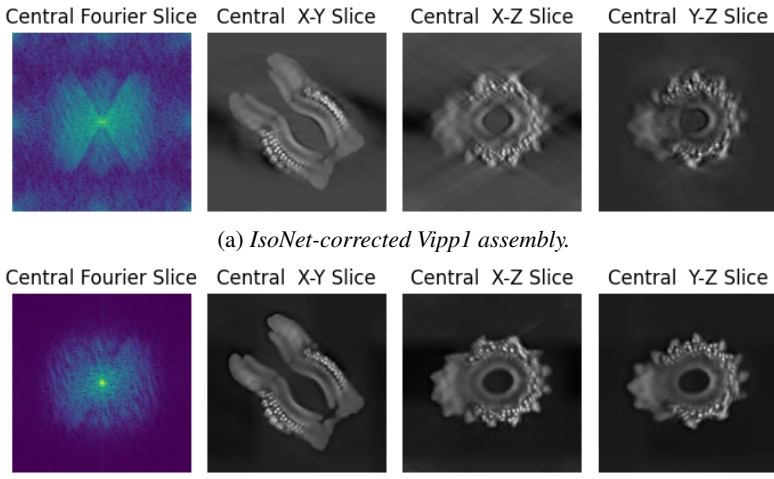

(a) *IsoNet-corrected Vipp1 assembly.*

(b) *CryoGEN-corrected Vipp1 assembly.*

Figure 8: *CryoGEN and IsoNet corrected Vipp1 assembly. The results shown in IsoNet exhibit strong streaking artifacts. In contrast, our method does not exhibit these streaking artifacts.*

Following the approach of IsoNet, we randomly rotated the tomograms to generate ten distinct samples before applying missing wedge corruption. As shown in Table 1, CryoGEN outperforms IsoNet, achieving higher PSNR and SSIM values. We present the original results in Figure 2, alongside spatial and Fourier domain comparisons in Figure 8. In the Fourier domain, CryoGEN accurately captures essential details and produces more consistent and symmetrical results. Furthermore, while IsoNet's reconstruction still exhibits artifacts, CryoGEN produces significantly smoother and more accurate results with higher contrast in spatial domain.

## 5.3 REAL-WORLD EXAMPLES

In this section, we evaluate the method's effectiveness using real-world examples. We employ a well-known Cryo-ET particle selection benchmark, specifically the dataset of purified ribosomes (Zhang et al., 2016), as well as the virus-like particle dataset of immature HIV-1 for both single-particle and tomographic reconstruction (Schur et al., 2016). CryoGEN significantly reduces the ringing effect and achieves higher contrast, while providing better compensation for the missing wedge compared to IsoNet's irregular distribution.

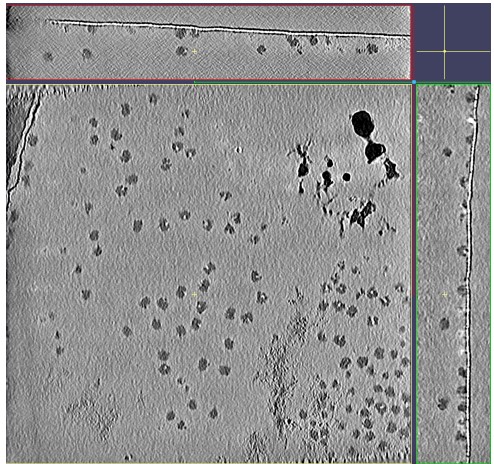

(a) *IsoNet-corrected purified ribosomes.*

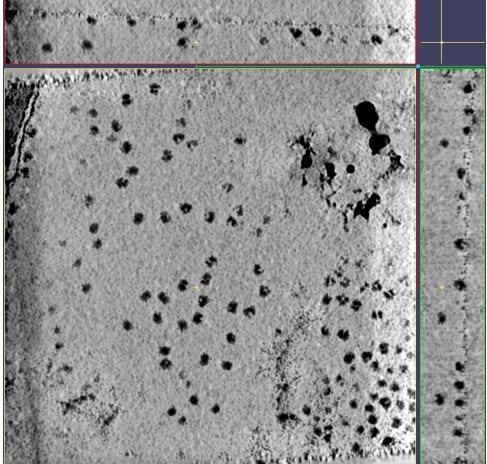

(c) *CryoGEN-corrected purified ribosomes.*

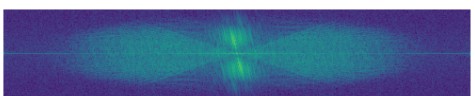

(b) *Central Fourier slice of the IsoNet-corrected purified ribosomes.*



(d) *Central Fourier slice of the CryoGEN-corrected purified ribosomes.*

Figure 9: *Comparison of IsoNet-corrected and CryoGEN-corrected purified ribosomes, including their corresponding central fourier slices. The CryoGEN-corrected images exhibit higher contrast and reduced high-frequency features. While both methods effectively fill in the missing wedge, the IsoNet correction shows an irregular distribution of high-frequency components in the central region, whereas CryoGEN achieves a more consistent distribution.*

**Purified Ribosomes.** The ribosomes dataset is commonly used as a Cryo-ET benchmark. We collected all seven tilt series from the EMPIAR-10045 dataset and applied the same preprocessing steps as IsoNet, detailed in Appendix A.8.5. Figure 9 shows the correction results for both IsoNet and CryoGEN. Ribosomes in the CryoGEN-corrected volume appear clearer and exhibit higher contrast, which significantly aids in particle selection. Additionally, there is less noise and fewer sharp artifacts in the background. Notably, the IsoNet-corrected volume displays a frequency spectrum with an irregular concentration of high-frequency components in the central region, introducing noticeable noise and artifacts. In contrast, the CryoGEN-corrected volume shows a much smoother and more consistent frequency distribution, with better control over central frequencies. The more symmetrical pattern suggests reduced distortion and better alignment with the expected smooth behavior, indicating improved data integrity.

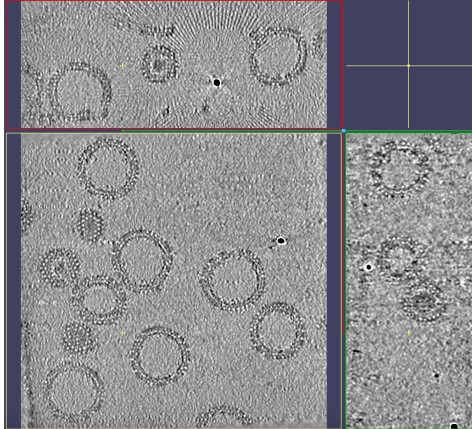

(a) *IsoNet-corrected immature HIV capsid.*

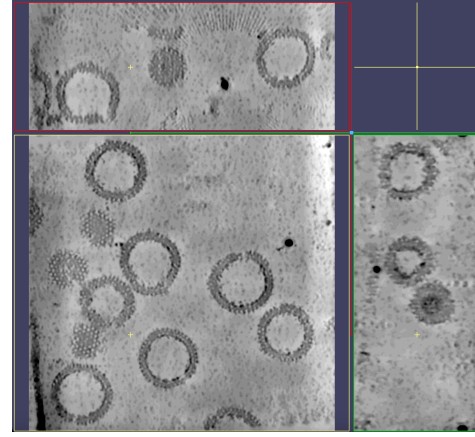

(c) *CryoGEN-corrected immature HIV capsid.*

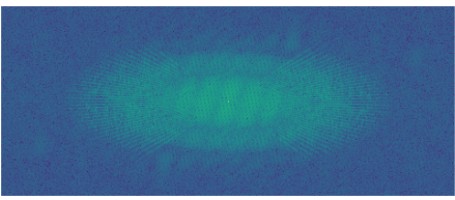

(b) *Central Fourier slice of the IsoNet-corrected immature HIV capsid.*



(d) *Central Fourier slice of the CryoGEN-corrected immature HIV capsid.*

Figure 10: *A comparison of IsoNet-corrected and CryoGEN-corrected immature HIV capsids, along with their corresponding central Fourier slices, reveals distinct differences. The IsoNet-corrected images exhibit noticeable ringing effects, whereas the CryoGEN-corrected images feature minimal noise and a smoother background. Additionally, while both methods address the missing wedge region, IsoNet correction introduces more prominent streak artifacts.*

**HIV Capsid.** The results of HIV capsid dataset are presented in Figure 10. Following IsoNet's pre-processing procedure, we collected three tilt series from the EMPIAR-10164 dataset and processed the volume as detailed in Appendix A.8.6. The CryoGEN-corrected HIV capsid is noticeably clearer than the IsoNet-corrected version, with minimal noise and a much smoother background. In contrast, the IsoNet-corrected volume exhibits a pronounced ringing effect around the gold beads, with bright rings surrounding them and unwanted white dust scattered throughout the image. Our algorithm effectively eliminates all these artefacts. Additionally, CryoGEN compensates for the missing wedge region more effectively than IsoNet. As shown in the top windows of Figure 10 (a) and Figure 10 (c), the virus particle in the CryoGEN-corrected volume is more intact, with fewer defects compared to the IsoNet-corrected version. In the Fourier domain, while both methods attempt to fill the missing wedge region, IsoNet's correction introduces more noticeable line artefacts.

## 6    CONCLUSION

Cryo-ET serves as an indispensable tool for visualizing subcellular architectures in their native environments, yet its potential is constrained by anisotropic resolution artifacts arising from the missing-wedge effect. In this study, we present CryoGEN, an innovative energy-based framework engineered to resolve the missing-wedge challenge in Cryo-ET. Our method achieves faster, more stable training convergence than existing approaches while significantly enhancing reconstruction fidelity.

ACKNOWLEDGMENT

The project was led by Dr. Teng, who oversaw its progression, designed the methodology, and implemented the algorithms. Guided by Dr. Teng during the internship, Yuxuan Ren refined the algorithms and performed the experimental validation. The authors would also like to extend their sincere gratitude to Prof. Zhouchen Lin and Dr. He Zhang for their exceptional feedback on this research, as well as Dr. Sha Hu for her assistance in refining the biological terms during the paper writing process. The authors also express the appreciation to Dr. Hui Wang and Dr. Jiayan Zhang for their motivational discussion and biological expertise.

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

# A  APPENDIX

## A.1  ALGORITHM FLOWCHART

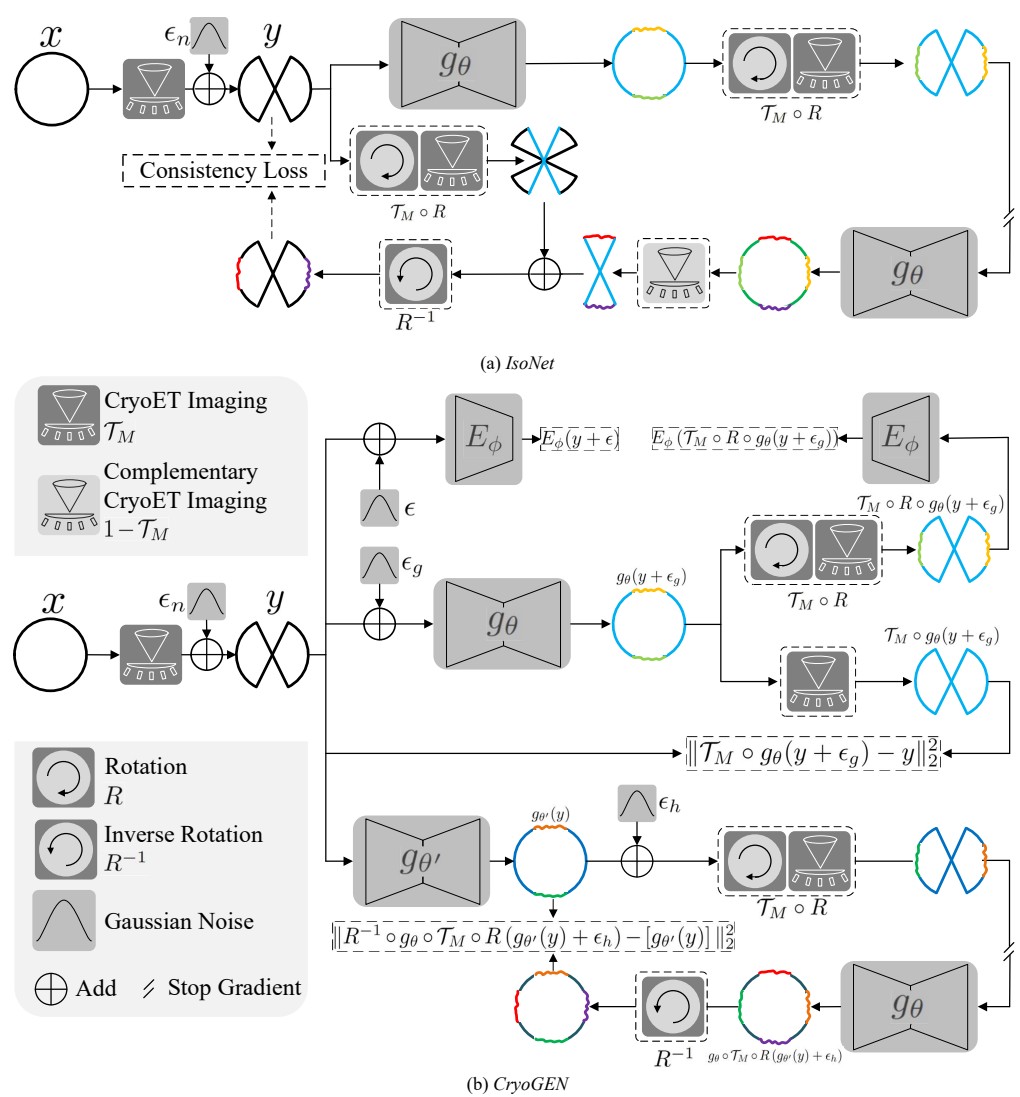

Figure 11: *Algorithm flowcharts of (a) IsoNet and (b) CryoGEN.*

IsoNet formulates the task of restoring $y$'s missing wedge information to reconstruct $x$ as an inpainting problem Lugmayr et al. (2022), solving it in a self-supervised manner as illustrated in Figure 11 (a). It trains a neural network, denoted as $g_\theta$, following these steps:

1. A sample $y$ is processed by $g_\theta$ to obtain a missing-wedge-restored $\tilde{x} = g_\theta(y)$.

2. The signal $\tilde{x}$ is rotated by a rotation operator $R$ (randomly chosen from a predefined set $\mathcal{R}$). Then, a simulated missing wedge operation $\mathcal{T}_M$ is applied, yielding $\tilde{y} = \mathcal{T}_M \circ R(\tilde{x})$.

3. The simulated sample $\tilde{y}$ is fed to $g_\theta$, yielding a reconstructed sample $\hat{x} = g_\theta(\tilde{y})$.

4. The inpainted part is first extracted as $\hat{x} - \mathcal{T}_M(\hat{x})$, then combined with the rotated $y$ with missing-wedge operation to obtain $\mathcal{T}_M \circ R(y)$, and finally the inverse rotation is applied to yield $\hat{y} = R^{-1}\left[(\hat{x} - \mathcal{T}_M(\hat{x})) + \mathcal{T}_M \circ R(y)\right]$.

5. $\hat{y}$ and $y$ form a data pair $(\hat{y}, y)$ to train $g_\theta$, where $y$ serves as the label.

6. These steps are repeated until convergence.

Similarly, details of CroGEN are provided in Figure Figure 11 (b), with the complete algorithm presented in Section 4.3.

## A.2 ROTATION LIST DEFINED IN THE ISONET

The cropped subtomograms are cube-shaped with six faces, resulting in 24 possible rotations for reorientation. However, we exclude the four rotations that maintain the same missing wedge in the $X$-$Z$ direction as the original, unrotated subtomogram. Further details and a schematic diagram are provided in the supplementary materials of IsoNet (Liu et al., 2022).

## A.3 ADDITIONAL RESULTS

### A.3.1 ADDITIONAL RESULTS FOR SECTION 5.1

Formally, $\hat{v}$ represents the predicted volume, and $v^*$ denotes the ground truth. PSNR and SSIM are defined as follows:

$$\text{PSNR} = 10 \log_{10} \frac{I_m}{\|\hat{v} - v^*\|^2}, \quad \text{SSIM}(\hat{v}, v^*) = \frac{(2\mu_{\hat{v}}\mu_{v^*} + C_1)(2\sigma_{\hat{v}v^*} + C_2)}{(\mu_{\hat{v}}^2 + \mu_{v^*}^2 + C_1)(\sigma_{\hat{v}}^2 + \sigma_{v^*}^2 + C_2)}.$$

In the PSNR formula, $I_m$ represents the maximum possible pixel value, which we define as the maximum value of the ground truth image.

In the SSIM formula:

- $\mu_{\hat{v}}$ and $\mu_{v^*}$ are the mean intensities of images $\hat{v}$ and $v^*$
- $\sigma_{\hat{v}}^2$ and $\sigma_{v^*}^2$ are the variances of images $\hat{v}$ and $v^*$
- $\sigma_{\hat{v}v^*}$ is the covariance between images $\hat{v}$ and $v^*$
- $C_1$ and $C_2$ are small constants used to stabilize the division when the denominator is close to zero

The central Fourier slices of the corrected sphere and prism are displayed in Figure 12, while the original clean sphere and prism are shown in Figure 13.

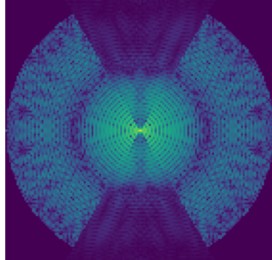

(a) Central fourier slice of the IsoNet-corrected sphere.

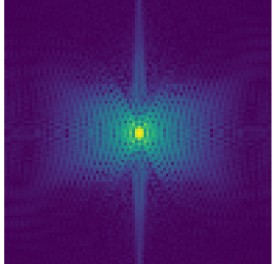

(b) Central fourier slice of the IsoNet-corrected prism.

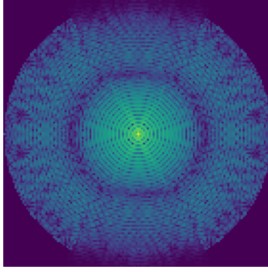

(c) Central fourier slice of the CryoGEN-corrected sphere.

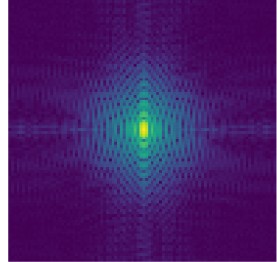

(d) Central fourier slice of the CryoGEN-corrected prism.

Figure 12: *Central fourier slices of sphere and prism. The IsoNet's corrected has less information both at low and high-frequency signals with missing regions, while CryoGEN fills in most of the missing wedge. It is consistent with the spatial domain results.*

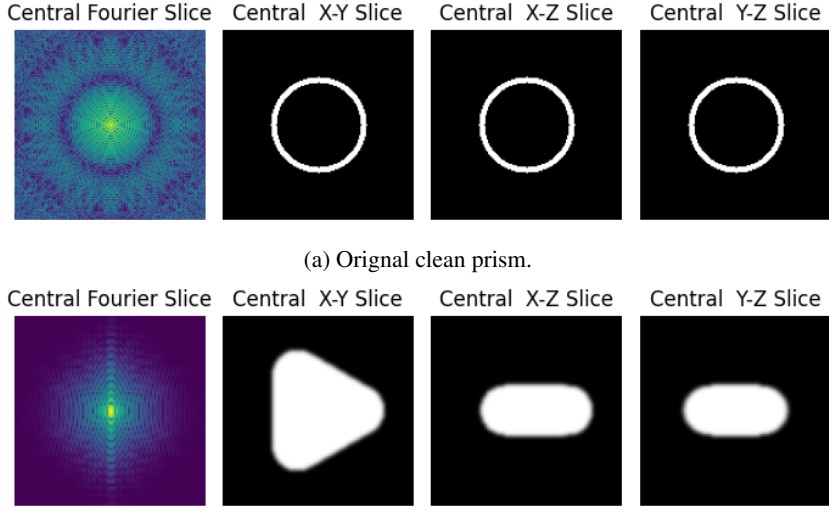

(a) Orignal clean prism.

(b) Original clean prism.

Figure 13: *Original clean shapes.*

Compared to the corrected tomograms, the CryoGEN-corrected versions more closely resemble the original clean shapes.

### A.3.2 ADDITIONAL RESULTS FOR SECTION 5.2

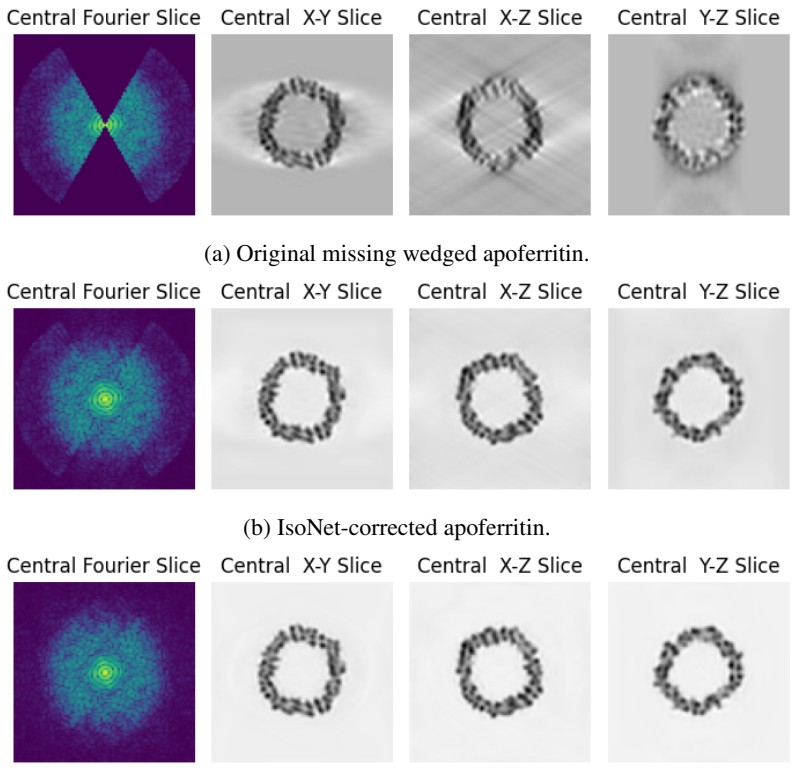

(a) Original missing wedged apoferritin.

(b) IsoNet-corrected apoferritin.

(c) CryoGEN-corrected apoferritin.

Figure 14: *Missing wedged, CryoGEN and Iso-Net corrected apoferritin.*

We present the central $X$-$Y$, $X$-$Z$, $Y$-$Z$ slices, as well as the central Fourier slices for the corrupted, IsoNet-corrected, and CryoGEN-corrected volumes in Figure 14. In the IsoNet-corrected $X$-$Y$ slice, there are faint white artifacts in the background, consistent with Figure 7, and line artifacts in the $X$-$Z$ slice, which are absent in the CryoGEN-corrected results. Additionally, the central Fourier slice of the IsoNet-corrected volume displays a distinct borderline, which is not present in the CryoGEN-corrected slice.

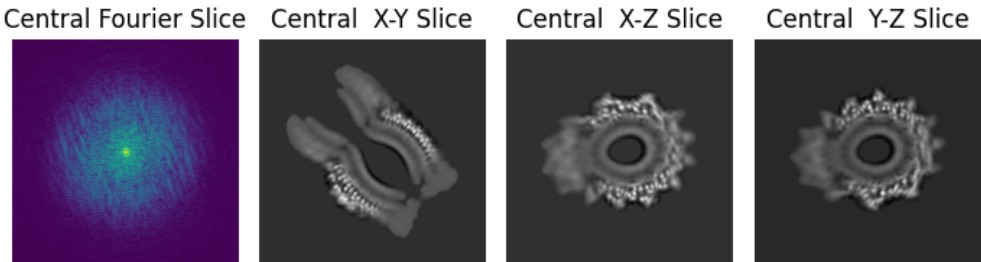

Figure 15: *Original clean Vipp1 assembly.*

The original clean Vipp1 assembly are shown in Figure 15.

### A.3.3 RECONSTRUCTION FROM NOISY SAMPLES

We also demonstrated the robustness of our algorithm under higher noise levels, such as SNR=0.2. The results are presented in Figure 16. Even in this challenging scenario, CryoGEN outperforms IsoNet, demonstrating its strong denoising capabilities.

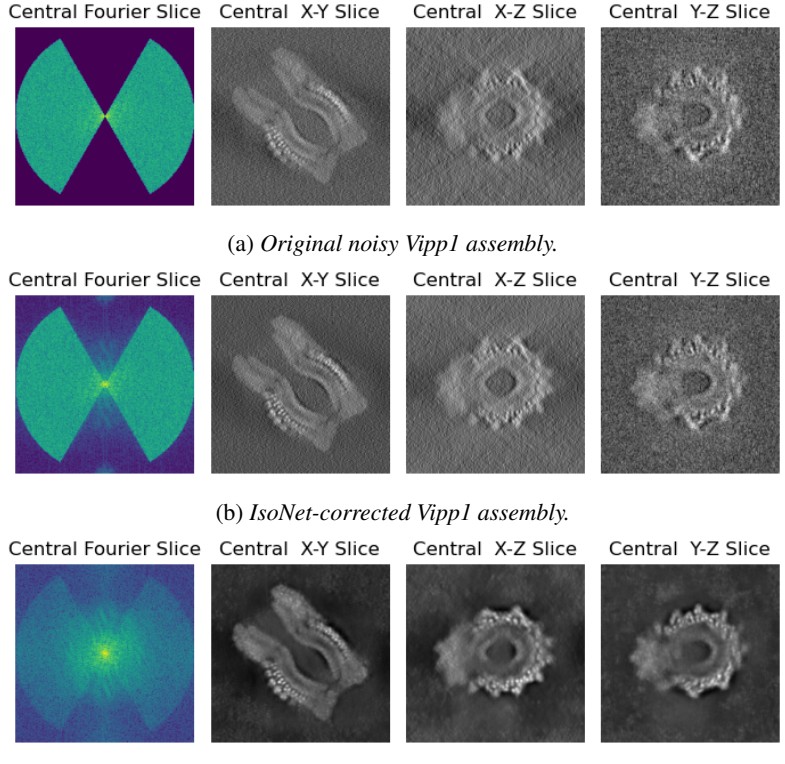

(a) *Original noisy Vipp1 assembly.*

(b) *IsoNet-corrected Vipp1 assembly.*

(c) *CryoGEN-corrected Vipp1 assembly.*

Figure 16: *CryoGEN and IsoNet corrected noisy Vipp1 assembly SNR=0.2.*

### A.4 Artifacts

In this section, we summarize the artifacts observed during our experiments, including *streak artifacts*, *synthetic artifacts*, and *ringing artifacts*. Throughout all experiments, CryoGEN consistently outperforms the baseline methods.

#### A.4.1 Streak Artifacts

*Streak artifacts* commonly appear around dense materials, such as *gold particles*, in Cryo-ET reconstructions, primarily due to two factors: (1) the restricted tilt angle range during data acquisition, and (2) the sensitivity of reconstruction algorithms, such as WBP (Radermacher, 2006), to high-intensity variations (see Figure 1). These bright spots generate ripples or streaks during the back-projection process, resulting in visual distortions that degrade the overall image quality.

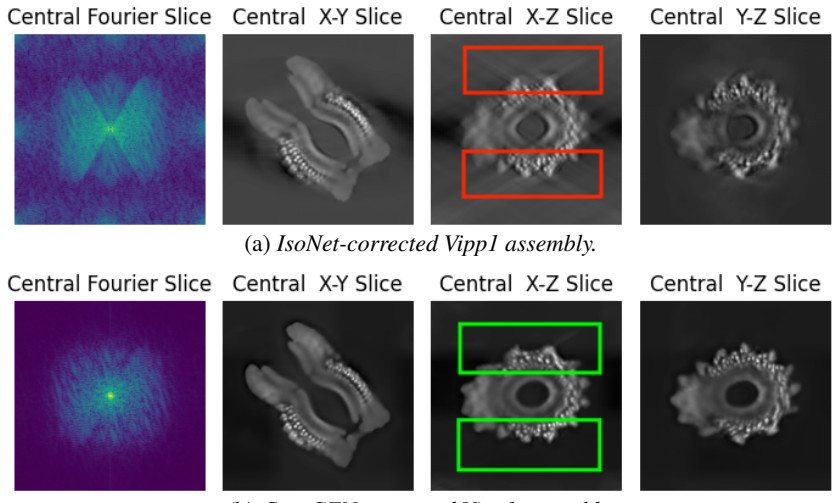

(a) *IsoNet-corrected Vipp1 assembly.*

(b) *CryoGEN-corrected Vipp1 assembly.*

Figure 17: *CryoGEN and IsoNet corrected Vipp1 assembly.*

Although these artifacts are clearly depicted in Figure 17, their visibility may be constrained by the size of the gold nanoparticles and ribosomes in Figure 9. To address this, we provide an additional view in Figure 18, where the artifacts are more noticeable.

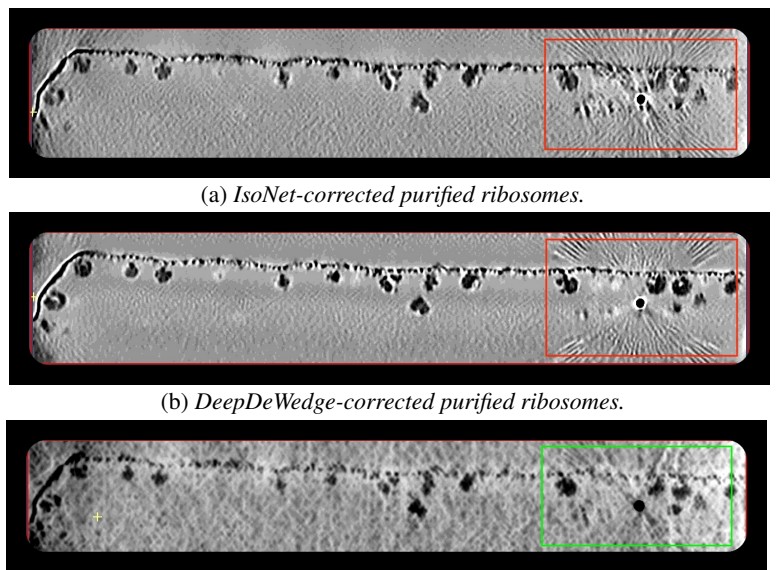

(a) *IsoNet-corrected purified ribosomes.*

(b) *DeepDeWedge-corrected purified ribosomes.*

(c) *CryoGEN-corrected purified ribosomes.*

Figure 18: *Comparison of IsoNet-corrected and CryoGEN-corrected purified ribosomes.*

### A.4.2 SYNTHETIC ARTIFACTS

*Synthetic artifacts* refer to unintended distortions or features that emerge in generated images or data, often resulting from generative models trained on datasets with *incomplete* distributions. As shown in Figure 19, the IsoNet method introduces synthetic artifacts caused by the reconstruction of the missing wedge, whereas the CryoGEN approach effectively resolves these issues.

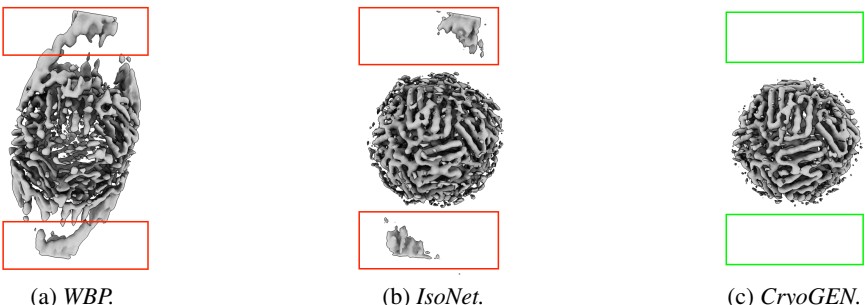

(a) *WBP.*   (b) *IsoNet.*   (c) *CryoGEN.*

Figure 19: *Comparison of low-density volumes generated by WBP (**a**), IsoNet (**b**) and CryoGEN (**c**).*

### A.4.3 RINGING ARTIFACTS

*Ringing artifacts* are undesirable distortions that appear as bands, typicallynear sharp edges in tomographic imaging. These artifacts often arise from reconstruction using band-limited signals that lack the high-frequency components required to accurately capture sharp transitions. In experiments with both purified ribosomes and immature HIV capsids, CryoGEN outperforms baseline methods, displaying minimal ringing artifacts.

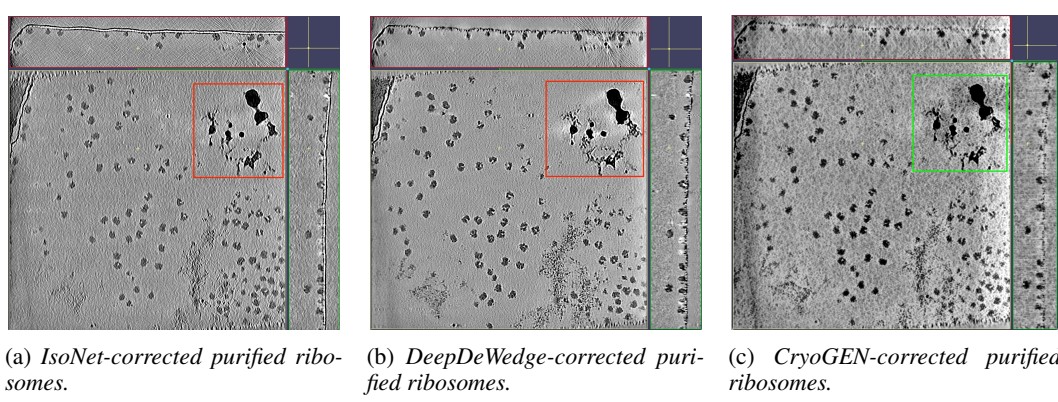

(a) *IsoNet-corrected purified ribosomes.*   (b) *DeepDeWedge-corrected purified ribosomes.*   (c) *CryoGEN-corrected purified ribosomes.*

Figure 20: *A comparison of purified ribosomes corrected by IsoNet, DeWedge, and CryoGEN.*

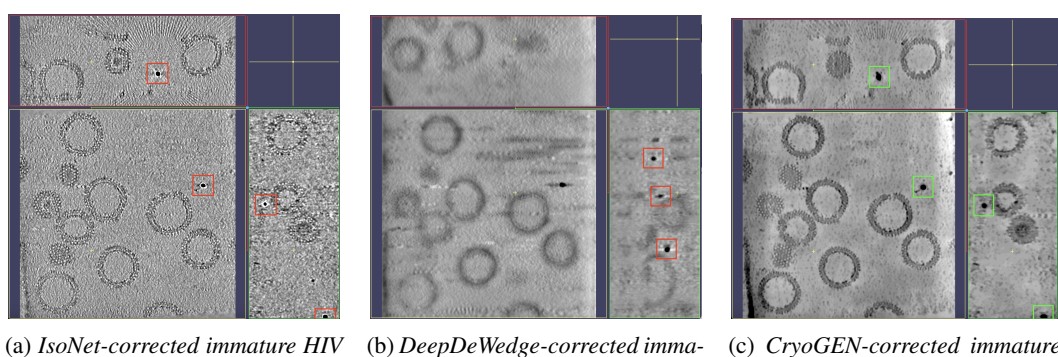

(a) *IsoNet-corrected immature HIV capsid.*   (b) *DeepDeWedge-corrected immature HIV capsid.*   (c) *CryoGEN-corrected immature HIV capsid.*

Figure 21: *A comparison of immature HIV capsids corrected by IsoNet, DeWedge, and CryoGEN.*

### A.5   ADDTIONAL RESULTS BY DEEPDEWEDGE

First, we present the DeepDeWedge-corrected simple shapes in Figure 22, where defects similar to those in the IsoNet-corrected versions are apparent. Both methods exhibit spatial domain artifacts, resulting in distorted volumes, and neither DeepDeWedge nor IsoNet effectively fills in the missing information in Fourier space.

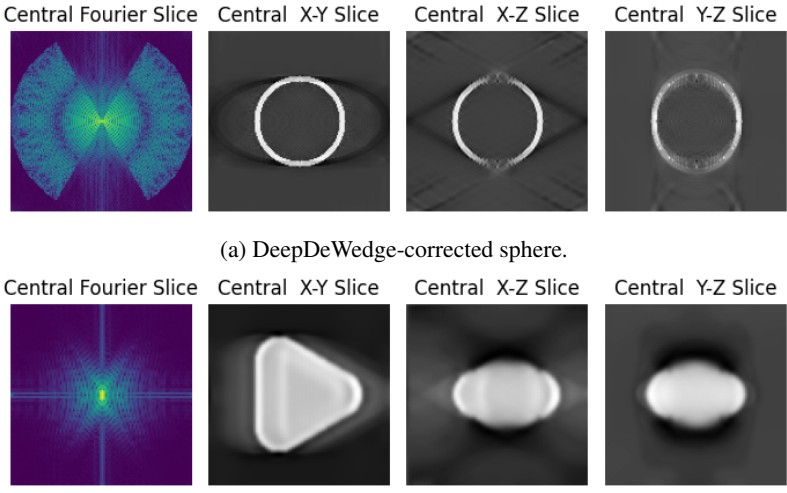

(a) DeepDeWedge-corrected sphere.

(b) DeepDeWedge-corrected prism.

Figure 22: *DeepDeWedge-corrected shapes.*

Next, the DeepDeWedge-corrected simulated data is displayed in Figure 23. Similar to IsoNet, DeepDeWedge encounters the same issues, generating faint shadows and distinct line artifacts in the background, as well as a noticeable borderline in the central Fourier slice.

Finally, we test the DeepDeWedge on the real-world examples as shown in Figure 24. The DeepDeWedge-corrected ribosomes exhibit the same irregular distribution of high-frequency components in the central region and yield unsatisfactory results for the HIV capsid, with noticeable artifacts. A potential reason for the poor performance on the HIV capsid may be the loss of information caused by even-odd splits.

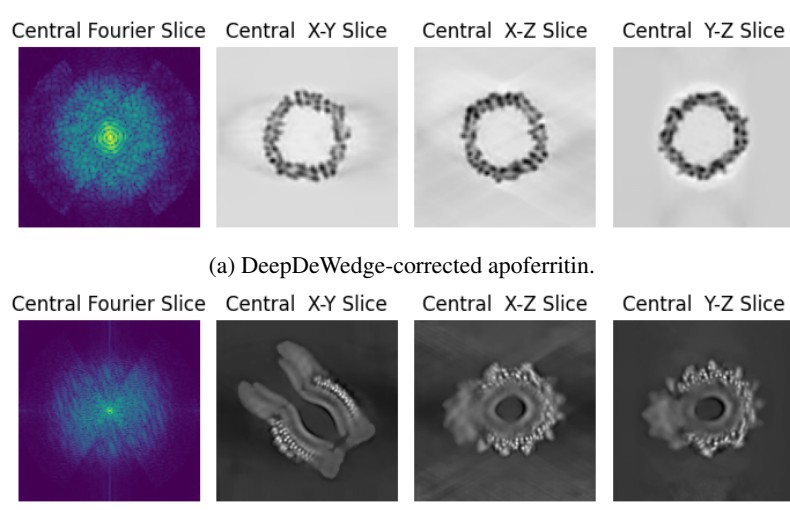

(a) DeepDeWedge-corrected apoferritin.

(b) DeepDeWedge-corrected Vipp1 assembly.

Figure 23: *DeepDeWedge-corrected simulated data.*

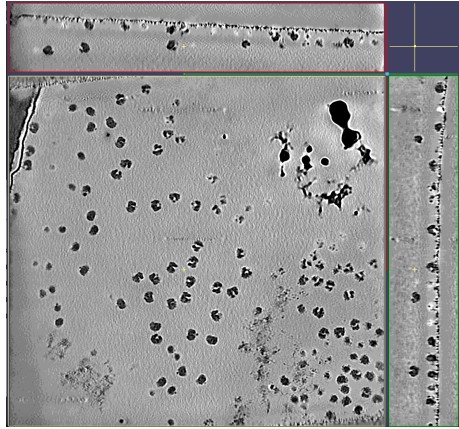

(a) DeepDeWedge-corrected purified ribosomes.

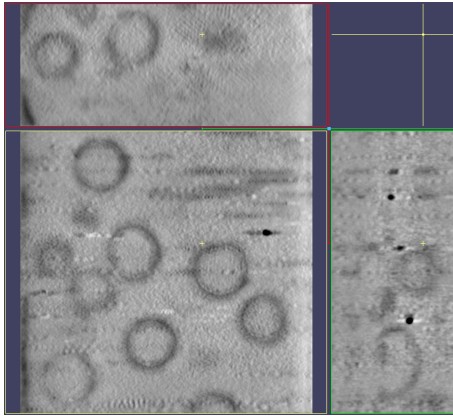

(c) DeepDeWedge-corrected HIV capsid.

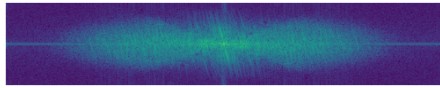

(b) Central fourier slice of the DeepDeWedge-corrected purified ribosomes.

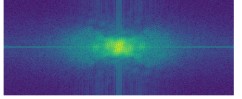

(d) Central fourier slice of the DeepDeWedge-corrected HIV capsid.

Figure 24: *DeepDeWedge-corrected ribosomes and HIV capsid, along with their corresponding central Fourier slices.*

### A.6 VISUALIZATION OF A SINGLE RIBOSOME

In Figure 25, we present a visualization of an individual ribosome. The average of all ribosomes is compared to those reconstructed by IsoNet and CryoGEN, using the fitmap command in ChimeraX[1] for rigid-body optimization to align the maps. While the IsoNet reconstructions may appear to exhibit more details in 2D grayscale images, as shown in Figure 9, this effect is likely due to undesired noise or synthetic artifacts, highlighting the superior denoising capabilities of CryoGEN.

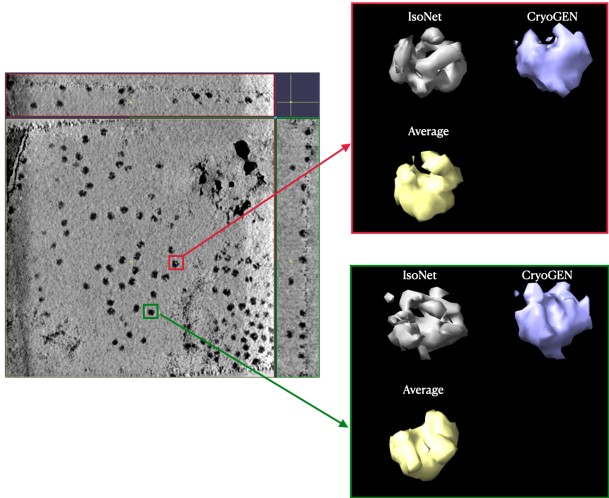

Figure 25: *Single Ribosome Visualization: We compared CryoGEN with IsoNet and the averaged results of all ribosomes across two examples at different rotation angles.*

Furthermore, we compute the correlation between the ribosome reconstructions generated by IsoNet and CryoGEN with the average, obtaining correlation coefficients of $\rho_{IsoNet}$ = 0.1792 and $\rho_{CryoGEN}$ = 0.5134, respectively ($\rho_{CryoGEN} > \rho_{IsoNet}$). Both qualitative and quantitative results demonstrate that CryoGEN outperforms IsoNet, providing smoother reconstructions.

### A.7 IMPLEMENTATION DETAILS

Following struct2map GAN (Zhang et al., 2024), the architecture consists of a generator and discriminator with specific design choices. The generator is a modified U-Net architecture, known as U-Net++ (Zhou et al., 2018), which enhances the standard U-Net with dense skip connections for improved performance. The discriminator is composed of four 3D convolutional layers, each using a 3×3×3 kernel. Following the final convolutional layer, an adaptive average pooling layer reduces the dimensions of the feature map to 1×1×1. This output is then flattened and fed into a series of three fully connected layers, with ReLU activations between each layer. The final layer produces a single output, which serves as the result of binary classification.

---

[1]https://cgl.ucsf.edu/chimerax/docs/user/commands/fitmap.html

The CryoGEN are trained with the Adam optimizer (Kingma & Ba, 2015) with batch size one for simulated shapes and protein subtomograms and with batch size $4$ for real-world examples. The learning rate is set to $0.0004$ with a linear warm-up phase in the initial one-tenth steps, which is followed by a linear decay schedule thereafter. Different from IsoNet, which progressively increases the noise scale, we apply random noise levels across all training steps. Specifically, a random number is sampled from a uniform distribution within the range $(0, 1]$ and multiplied by the set noise scale for each step. Additionally, the penalty term $\lambda$ is kept constant during the first epoch and then decays linearly throughout the subsequent epochs.

### A.8  EXPERIMENT DETAILS

#### A.8.1  SPHERE

A hollow sphere with an outer diameter of $70$ pixels and a thickness of $4$ pixels is positioned at the center of a $140{\times}140{\times}140$ volume. Corruption is applied by setting values to zero within the missing wedge angles in Fourier space. For training, the volume is split into ten $96{\times}96{\times}96$ pixel subtomograms with randomly chosen origins. These subtomograms are then randomly cropped to $64{\times}64{\times}64$ pixels before being input into the models.

#### A.8.2  PRISM

A prism with a thickness of $20$ pixels is placed inside a $96{\times}96{\times}96$ volume. It is randomly rotated in ten directions. Corrupted prisms are generated by setting zero values within the missing wedge angles in Fourier space. The entire volume is directly fed into the model during training.

#### A.8.3  SIMULATED APOFERRITIN

Ten randomly rotated apoferritin datasets are downloaded from a link provided by IsoNet and generated using ChimeraX's molmap function. During training, the datasets are directly fed into the model without further modifications.

#### A.8.4  SIMULATED STACKED RINGS

C13 Vipp1 stacked ring data are downloaded from the EMDB database and binned twice, resulting in $200{\times}200{\times}200$ pixels. The data is randomly rotated in ten different directions. Corrupted stacked rings are generated by setting zero values within the missing wedge angles in Fourier space. For training, the data is split into ten $96{\times}96{\times}96$ pixel subtomograms with random zero origins, then randomly cropped to $64{\times}64{\times}64$ pixels before being fed into the models.

#### A.8.5  RIBOSOMES

Ribosome data is downloaded from the EMPIAR database and binned six times, yielding a pixel size of $13.02$ Å. IsoNet's deconvolution is applied, following the same procedure as described by (Liu et al., 2022). To ensure that subtomograms contain sufficient data, IsoNet's mask generation tool is used to extract subtomograms with at least $40\%$ non-zero pixels based on the density mask. For training, a tomogram is split into seventy $80{\times}80{\times}80$ pixel subtomograms, resulting in a total of $490$ subtomograms. These are randomly cropped to $64{\times}64{\times}64$ pixels before being fed into the models.

#### A.8.6  HIV CAPSID

Raw tilt series for the HIV capsid is downloaded from the EMPIAR database. The movie stacks are drift-corrected and reconstructed using the WBP algorithm, aided by the latest tomogram processing tools such as Aretomo2. The processed tomograms, TS-01, TS-43, and TS-45, are then deconvolved following the procedure outlined by (Liu et al., 2022). IsoNet's mask generation tool is applied to ensure that each subtomogram contains at least $50\%$ non-zero pixels. During training, each tomogram is split into one hundred $96{\times}96{\times}96$ pixel subtomograms, resulting in 300 subtomograms. These are randomly cropped to $64{\times}64{\times}64$ pixels before being fed into the models.

