# OpenReview forum: "CryoGEN: Generative Energy-based Models for Cryogenic Electron Tomography Reconstruction"
_ICLR.cc/2025/Conference — ICLR 2025 Poster_

### Official Review · Reviewer_wTW8 · 2024-10-30

**Soundness:** 3
**Presentation:** 3
**Contribution:** 3
**Rating:** 8
**Confidence:** 3

**Summary:**

CryoGEN's author(s) proposed a new method called CryoGEN, aimed at addressing the "missing wedge" problem and the challenge of low signal-to-noise ratio in cryo-electron tomography (Cryo-ET). CryoGEN combines generative adversarial networks (GANs) with energy-based models to produce more consistent and high-quality 3D reconstructions. Compared to traditional weighted back-projection (WBP) and existing deep learning methods like IsoNet, CryoGEN effectively fills in missing information and reduces blurring artifacts.

Key contributions :

1. The introduction of CryoGEN, which uses energy-based models to handle multiple possible solutions, avoiding the blurring effect caused by simple averaging.

2. The inclusion of consistency loss and noise injection to ensure that generated images maintain fidelity to the original data while preserving diversity in the results.

3. Experimental validation on multiple datasets, such as HIV viral particles and neural synapses, where CryoGEN outperforms existing baseline methods in metrics like peak signal-to-noise ratio (PSNR), structural similarity index (SSIM), and Fourier shell correlation (FSC).

In all, the authors announces a more effective solution for reconstruction in Cryo-ET, showing significant improvements in quality and stability. Future work includes improving training efficiency, automating parameter adjustments, and extending its applications to other cryo-electron microscopy domains.

**Strengths:**

Originality: The originality of the paper lies in its attempt to improve and combine existing methods by using generative adversarial networks (GANs) and energy-based models to address the missing wedge problem in cryo-electron tomography (Cryo-ET).

Quality: The quality of the research is validated through experiments on multiple datasets, including synthetic samples and real biological samples (e.g., HIV viral particles and neural synapses). However, the experimental design lacks in-depth comparison with other state-of-the-art methods, especially the absence of sufficient ablation studies to demonstrate the importance of each model component.

Clarity: The paper is generally well-written with a logical flow, but some parts are explained too briefly, particularly the mathematical description and implementation details of the energy model, which might make it difficult for non-expert readers to fully understand the working principles.

Significance: The potential impact of CryoGEN in the field of cryo-electron tomography is limited. Although the method improves reconstruction quality to some extent, the degree of improvement is relatively modest, and challenges may arise when applied to more complex datasets.

**Weaknesses:**

Lack of Novelty: Although CryoGEN combined GANs and energy-based models, this architecture still relies on existing methods and lacks truly groundbreaking innovation. For instance, the combination of GANs and energy-based models has already been widely applied in other fields, and the paper does not sufficiently demonstrate the unique contribution of this combination in Cryo-ET. We recommend citing some recent literature, such as "Your GAN is Secretly an Energy-based Model and You Should Use It" by Tong Che et al. (2020), which discusses the connection between GANs and energy-based models and their effectiveness in data reconstruction. Furthermore, it is suggested to elaborate on the specific aspects where this combination lacks novelty in Cryo-ET applications, such as similarities in loss functions, model architecture redundancy, or training strategy limitations, and clarify how CryoGEN distinguishes itself from existing methods to enhance the contribution statement of the paper.

Incomplete Theoretical Explanation: The mathematical description and implementation details of the energy model are overly simplified. It is recommended to provide a more detailed description of the energy model's training process, including key equations and their roles in optimizing the model. Adding intuitive examples or visual aids, such as diagrams illustrating the training dynamics of the energy model, would make the content more accessible to non-expert readers. Specifically, expanding on equations (3) and (4) would help clarify how the energy model interacts with the GAN during training. These additional details would help address potential confusion and make the feedback more actionable.

Lack of In-Depth Result Analysis: Although the experimental results show performance improvement, the authors do not provide in-depth discussion on the reasons behind these key performance improvements. It is recommended to provide a more detailed analysis of why CryoGEN performs better on specific datasets compared to other methods, which would better support the validity of its contributions.

**Questions:**

1. Lack of Comparative Experiments: It is suggested to add comparisons with other recent methods, particularly with more challenging and complex models, to better validate the advantages of CryoGEN.

2. Supplemental Theoretical Explanation: Could the authors provide more detailed explanations regarding the implementation details of the energy model? Adding intuitive examples and diagrams would help readers understand its complexity.

---

> ### Author Response · Authors · 2024-11-21
> **To Reviewer wTW8**
>
> We sincerely thank the reviewer for their positive evaluation and recognition of our work's contributions. We apologize for any misunderstandings and would like to take the opportunity of rebuttal to clarify the strengths of our work.
>
> 1. **Contribution and Originality:**
>     * To the best of our knowledge, we are the first to propose a comprehensive solution to the missing wedge problem in CryoET using Bayes' rule. Unlike most CryoEM algorithms, which often rely on known prior distributions, our approach addresses this as an unsupervised learning problem. This introduces additional complexity to the mathematical formulation.
>
>     * Our identification of IsoNet's limitations is unique, emerging from a non-convex optimization perspective. The primary contribution of our work lies in the mathematical formulation and the algorithmic framework. While we employ standard techniques from the field of computer vision, this should be seen as a strength rather than a limitation, as these techniques verify the generalizability of our approach.
>
> 2. **Theoretical Analysis:**
>     Formally analyzing CryoET from a theoretical perspective presents inherent challenges. While a comprehensive theoretical analysis is beyond the scope of this work, we offer key insights: although GANs are susceptible to mode collapse, they are highly effective in addressing mode averaging issues, as discussed in the motivation section. This is a primary reason for their inclusion in our approach. In the final version, we will provide a more detailed explanation and explore potential theoretical analyses.
>
> 3. **In-Depth Analysis:**
>     Section 3 provides an illustrative analysis of why CryoGEN outperforms IsoNet, with Figures 5 and 6 highlighting the core motivations. Additional experiments have been conducted and included in the revised version. Figure 16 demonstrates CryoGEN's enhanced denoising capabilities. Furthermore, the comparison indicates that IsoNet consistently suffers from residual effects of the missing wedge problem across all experiments, while CryoGEN effectively mitigates this issue.
>
> 4. **Comparison to SOTA methods:**
>     We have compared our method with the state-of-the-art (SOTA) methods, IsoNet and DeepDeWedge, the latter being published shortly before our submission. To the best of our knowledge, these represent the current SOTA. However, we are open to conducting additional experiments if the reviewer can suggest other relevant methods.
>
> 5. **Large Datasets and Deep Models:**
>     Compared to the SOTA methods, CryoGEN exhibits greater robustness when applied to large datasets and deep models. This observation stems from IsoNet’s need for meticulous fine-tuning at each iteration. A plausible explanation is that IsoNet’s recursive training approach may accumulate errors over iterations, leading to instability. In contrast, our algorithm avoids recursive updates, ensuring faster training and enhanced stability. Furthermore, our experiments spanned a range of difficulty levels, from straightforward cases to the most challenging scenario involving the HIV ribosome.
>
> 6. **Significance of CryoGEN for CryoET:**
>     We utilize the isotropy property of CryoET, assuming that distributions are uniform across all orientations. To the best of our knowledge, this is a novel approach that has not been formally proposed in the CryoET field before. Furthermore, we are the first to formally define and address the missing-wedge problem, a longstanding challenge in CryoET, with a consistent solution.

---

> > ### Comment · Reviewer_wTW8 · 2024-11-28
> > **To authors**
> >
> > Thank you for your detailed and thoughtful rebuttal. I appreciate the effort you put into addressing our concerns and further clarifying the contributions of CryoGEN. Your explanations regarding the innovative aspects, particularly the use of Bayesian unsupervised learning and the emphasis on IsoNet’s limitations, are compelling. Below, I would like to provide additional suggestions and raise some follow-up questions to further improve the clarity and impact of your work.
> >
> > 1.	Complex Dataset Experiments
> >
> > While your experiments on datasets such as HIV viral particles and neural synapses are convincing, I suggest incorporating experiments on more complex datasets with non-uniform distribution characteristics. This would provide stronger evidence of CryoGEN’s robustness and generalizability across diverse scenarios. For instance, such datasets might include irregularly shaped particles or structures with significant variation in density.
> >
> > 2.	Systematic and Intuitive Experiment Design
> >
> > I appreciate the additional analysis and figures (e.g., Figure 16) provided in your rebuttal. To further enhance the presentation, I recommend including more intuitive diagrams and visual aids in the final version. Specifically, clear visualizations of the energy model and GAN interaction mechanisms, as well as an illustration of training dynamics, could significantly aid in understanding the interplay between these components. Additionally, expanding the comparative analysis with state-of-the-art methods, both quantitatively and qualitatively, would strengthen the systematic evaluation of your approach.
> >
> > 3.	Choice of Generative Model: GAN vs. Diffusion Models
> >
> > Your rebuttal highlights the advantages of GANs, particularly in addressing mode averaging and achieving faster training. However, given recent advances in diffusion models, which are highly effective for high-resolution image generation and handling noise, I wonder if you considered and evaluated diffusion models during your design process. These models could potentially offer advantages for Cryo-ET, especially in mitigating noise and filling in missing information. A brief discussion of the rationale behind selecting GANs over other generative approaches, including diffusion models, would add depth to the justification of your methodology.

---

> > > ### Author Response · Authors · 2024-12-03
> > >
> > > Thank you once again for raising these questions and helping us enhance the quality of our work. We will strive to incorporate these suggestions into our final version. In particular, diffusion models will be part of our future work, and we will include a clear discussion on this topic as well.

---

### Official Review · Reviewer_gbuV · 2024-10-30

**Soundness:** 3
**Presentation:** 3
**Contribution:** 3
**Rating:** 6
**Confidence:** 3

**Summary:**

The paper addresses the limitations of Cryogenic Electron Tomography (Cryo-ET) in reconstructing 3D structures of cellular components due to the missing wedge problem, which creates anisotropic resolution in tomograms. The authors introduce CryoGEN, a generative energy-based model designed to tackle the missing wedge problem more effectively and stably, without requiring recursive prediction.

**Strengths:**

1. The use of energy-based models for addressing the missing wedge problem is novel and shows promise in improving reconstruction quality.
2. CryoGenic is computationally efficient, achieving significant runtime reductions compared to IsoNet, which would be valuable in large-scale biological studies
3. The paper is well-written and easy to follow

**Weaknesses:**

1. Couldn't find big technical issues.
2. A more detailed discussion would be needed (e.g., limitations, what would be the future work).
3. Typo: Page 1, line 32 – Repeated phrase "insights insights into."

**Questions:**

1. What is the SNR of the data shown in Figure 7? I’m curious to see 3D reconstruction results (comparison with IsoNet) with higher noise (SNR 0.01 or 0.001) levels.
2. On page 6: "At the inference stage, we begin by cropping the complete tomogram into multiple overlapping subtomograms" – Could you specify the number of crops used?

**Details Of Ethics Concerns:**

No concerns

---

> ### Author Response · Authors · 2024-11-21
> **To Reviewer gbuV**
>
> We thank the reviewer for acknowledging our contributions and offering positive feedback. We will address the reviews raised by the reviewers accordingly.
>
> 1. **Discussion:** We have partially addressed the limitations and potential future directions in our previous responses, and we will integrate these discussions into the revised version.
>
> 2. **Typos:** The identified typos have been corrected. We appreciate your attention to detail.
>
> 3. **Number of Crops:** We employ the same method proposed in Isonet. For the ribosome, the number of crops varies based on the input size, as the crop size is fixed at 96 for each dimension. Detailed training procedures are provided in Appendix A.6.
>
> 4. **SNR of the data shown in Figure 7:** The SNR can be considered effectively infinite since no noise is added. When the SNR is 0.01 or 0.001, the figures are too corrupted to contain any useful information, making learning impossible for all methods. However, we have conducted additional experiments with SNR = 0.2, and the updated results are provided in Appendix A.3.3.

---

### Official Review · Reviewer_PzBU · 2024-11-03

**Soundness:** 3
**Presentation:** 3
**Contribution:** 2
**Rating:** 6
**Confidence:** 3

**Summary:**

This paper introduces a new method for reconstructing 3D structures from cryo-electron tomography (Cryo-ET) data. It aims to address a common issue in Cryo-ET reconstruction known as the missing wedge problem. The proposed algorithm, CryoGEN, tackles this by using an energy-based model that learns to generate realistic 3D structures from incomplete information. The model includes two main components: an energy model, which scores how realistic a structure is, and a prediction model, which reconstructs the missing regions. CryoGEN also adds noise to the input data during training to handle the challenges posed by the one-to-many mapping issue. This approach enables the model to create accurate and clear reconstructions by filling in missing areas based on learned patterns from data. The paper uses four datasets to evaluate the effectiveness of CryoGEN, particularly in comparison to the baseline solution, IsoNet. Results show that CryoGEN produces clearer and more complete reconstructions with better preservation of structural details. On three of the datasets, quantitative evaluation is performed, demonstrating that CryoGEN surpasses IsoNet.

**Strengths:**

1. This paper addresses the missing wedge problem, a major issue in Cryo-ET that causes incomplete and blurred 3D reconstructions.

2. The paper proposes a novel deep learning solution for Cryo-ET reconstruction. Using GANs and energy models is a reasonable approach.

3. Diverse datasets are considered for evaluation, including both simulated and real tomograms.

**Weaknesses:**

1. The techniques used in CryoGEN are fairly standard in general computer vision. There doesn’t appear to be any part of the algorithm specifically motivated by the unique characteristics of Cryo-ET data, such as the missing wedge. For example, the motivation discussed in Section 3 is very common in general generative models like VAE and EBM. Also, EBM and GAN models have been widely used in general inpainting problems but not discussed as related work.

2. It's unclear why Equation (8) represents the posterior distribution. Is Equation (8) an implementation of Equation (3)?

3. This paper improves upon IsoNet by incorporating the energy model and noise perturbation on the input. Which of these additions is more essential to the strong performance of CryoGEN?

4. Regarding computational complexity, the paper provides inconsistent information. Section 5.3 reports that CryoGEN is much faster than IsoNet. However, CryoGEN is clearly more complex than IsoNet in terms of architecture. The authors also mention that the energy model converges more slowly in Section 4.1.

5. As mentioned in related work, DeepDeWedge can be used to address denoising and missing wedges simultaneously. Why is DeepDeWedge not included in the quantitative evaluation?

**Questions:**

See Weaknesses

---

> ### Author Response · Authors · 2024-11-15
> **To Reviewer PzBU - Part 1**
>
> We thank the reviewer for the detailed comments and helpful feedback. We also appreciate the acknowledgment of our contribution. We will address each of your remarks individually and hope to clarify any concerns.
>
> 1. **Contribution and Originality:**
>
>     * To the best of our knowledge, we are the first to provide a comprehensive solution for the missing wedge problem in CryoET with Bayes rule. It brought to our attention that our approach was developed concurrently with the work *CryoFM*. However, Unlike *CryoFM*, which relies on CryoEM data with complete information to construct $p_x(x)$, we build $p_x(x)$ solely from the distribution of incomplete data $p_y$. This introduces additional complexity into the mathematical formulation.
>
>     * Specifically, we leverage the **isotropy** property of CryoET: after constructing $p_y$, we define $\log p_x(x) \propto \frac{1}{|\mathcal{R}|} \sum_{R \in \mathcal{R}} \log p_y(\mathcal{T}_M \circ R \circ x)$. This rotational operation assumes that the distributions are uniform across all directions. To our knowledge, this represents a novel contribution that has not been previously proposed.
>
>     * Furthermore, inpainting algorithms typically rely on stronger assumptions compared to ours. For instance, [1] trains its model on a dataset of complete images, which aligns with our initial observation where $p_x(x)$ is assumed to be known. However, this assumption is generally not valid in CryoET. We will highlight this distinction in our revised paper.
>
> 2. **Motivation:**
>
>     The motivation discussed in Section 3 may have been addressed in general generative models but likely from different perspectives and aimed at solving distinct challenges:
>
>     * For example, while GANs are known to suffer from mode collapse issues, they are also effective in addressing problems related to mode averaging. This strength is one of the reasons we incorporate GANs into our approach, although it is not the only solution.
>
>     * In contrast, VAEs are prone to averaging issues due to their smoother latent space representation and the regularization inherent in their formulation. Specifically, the KL divergence term in the VAE objective encourages the posterior $q(z|x)$ to stay close to the prior.
>
>     * Finally,  our identification of IsoNet's limitations is unique, emerging from a non-convex optimization perspective. Our motivation complements these existing approaches while addressing challenges specific to our context. The primary contribution of our work lies in the mathematical formulation and the algorithmic framework. While we employ standard techniques from the field of computer vision, this should be seen as a strength rather than a limitation, as these techniques verify the generalizability of our approach.
>
> 3. **Equation 8 is an implementation of Equation 3**
>
>     Yes, you are correct. Specifically, the order of the likelihood and prior is reversed between Equation 8 and Equation 3, and we will fix this in the revision. Thanks for pointing out.
>
> 4. **Both the incorporation of an energy model and noise perturbation to the input are essential components**
>    * *Energy Model*: The energy model serves as a fundamental component in the procedure for estimating the prior $p_x(x)$ from $p_y(y)$. This process involves the following steps:
>
>       * Defining the Energy Function $E(y)$: The goal is to construct an energy function where regions close to data points in $\mathcal{Y} = {y_i}$ have low energy, and regions far from the data points have high energy. To achieve this, we leverage GANs (refer to [2] for details) to assign low $E$ values to high-density regions and high $E$ values to low-density regions.
>
>       * Constructing $p_y$: Once the energy model $E$ is trained, we construct $p_y$ using an energy-based representation, where $p_y(y) \propto \exp(-E(y))$, consistent with the Boltzmann distribution.
>
>       * Constructing $p_x$: Using the isotropy property of CryoET, we define $\log p_x(x) \propto \frac{1}{|\mathcal{R}|} \sum_{R \in \mathcal{R}} \log p_y(\mathcal{T}_M \circ R \circ x)$, where $R$ denotes the rotation operation and $\mathcal{T}_M$ represents the application of the missing-wedge effect.
>
>       By integrating these steps, we obtain $p_x(x)$, demonstrating the critical role of the energy model in our approach.
>
>    * *Noise Perturbation*: As highlighted in the motivation section, if the cardinality of $X|y = \arg\max p(x|y)$ is greater than one, defining a parameterized function $g_\theta$ that maps a single observation to all possible values of $x$ becomes infeasible, resulting in a one-to-many mapping. To address this, we add $y + \epsilon$, where $\epsilon$ has full support over the real domain, effectively providing infinite cardinality and transforming the problem into a many-to-fewer mapping. This approach not only resolves the mapping issue but also enhances training stability, as discussed in [3].

---

> ### Author Response · Authors · 2024-11-15
> **To Reviewer PzBU - Part 2**
>
> 5. **Computational Efficiency**
>
>     We apologize for any confusion regarding computational efficiency. To clarify, our approach is faster during training while maintaining the same computational complexity during inference:
>
>    * *Training*: IsoNet employs a recursive training process, whereas our method does not. This design choice makes our algorithm not only more robust during training but also over $\mathbf{\times 10}$ faster to converge. Moreover, IsoNet requires precise fine-tuning at each iteration, and its recursive nature can result in error accumulation and instability. In contrast, our approach avoids recursive updates, enabling faster training and greater stability.
>
>    * *Inference*: For the prediction model, we use an architecture with the same computational complexity as IsoNet, ensuring that the time complexity during inference remains similar.
>
> 6. **DeepDeWedge Results**
>
>     Apologies for the omission of results. DeepDeWedge was published shortly before our submission, but we have included XYZ slice images generated by DeepDeWedge in the appendix. We present the quantitative results below and they have been incorporated into the revised paper. Moreover, DeepDeWedge does not alter the formulation and uses IsoNet as a backbone, thereby inheriting all of its drawbacks.
>
>     -------------------------------------------------------------------------------------------------
>
>     *Table: Quantitative evaluation of image quality for tomography reconstructions using different methods, comparing PSNR and SSIM metrics (higher values indicate better performance for both metrics) on sphere, prism, and Vipp1 assembly datasets.*
>
>     | **Data State**        | **Sphere PSNR** | **Sphere SSIM** | **Prism PSNR** | **Prism SSIM** | **Vipp1 Assembly PSNR** | **Vipp1 Assembly SSIM** |
>     |------------------------|-----------------|-----------------|----------------|----------------|--------------------------|--------------------------|
>     | **Corrupted**          | 21.12           | 0.8113          | 14.82          | 0.6931         | 26.68                   | 0.8000                   |
>     | **Iso-corrected**      | 22.98           | 0.8770          | 19.11          | 0.8857         | 27.12                   | 0.8191                   |
>     | **Dewedge-corrected**  | 23.17           | 0.8824          | 21.10          | 0.9278         | 28.75                   | 0.8758                   |
>     | **CryoGEN-corrected**  | **29.19**       | **0.9706**      | **32.69**      | **0.9949**     | **30.65**               | **0.9199**               |
>
> ---
> [1] Lugmayr, Andreas et al. “RePaint: Inpainting using Denoising Diffusion Probabilistic Models.” *2022 IEEE/CVF Conference on Computer Vision and Pattern Recognition (CVPR)* (2022): 11451-11461.
>
> [2] Murphy, Kevin Patrick. *Probabilistic Machine Learning: Advanced Topics*. MIT Press, 2023. Chapters 24 and 26.
>
> [3] Arjovsky, Martin, and Léon Bottou. "Towards Principled Methods for Training Generative Adversarial Networks." *International Conference on Learning Representations*, 2017, https://openreview.net/forum?id=Hk4_qw5xe.

---

> ### Comment · Reviewer_PzBU · 2024-11-29
>
> Thank you for the clarification. The authors' responses have addressed most of my concerns. So I changed my score from 5 to 6. While the motivation presented in Section 3 still appears very general to me, I am largely convinced of the technical novelty of the proposed method. One remaining confusion is that, why is "leveraging the isotropy property of CryoET" considered a specific technical contribution? Also I suggest that the authors include a discussion of image inpainting approaches in the related work section, as these share similar problem settings and techniques with cryo-ET reconstruction.

---

> > ### Author Response · Authors · 2024-12-03
> >
> > Leveraging the isotropy property of CryoET is not one of our contributions; rather, it is a necessary condition for implementing self-supervised learning. We will clarify this point in the revised version. We will also address the in-painting approach as recommended.
> >
> > Once again, thank you for taking the time to review our response. We truly appreciate your feedback, suggestions, and recognition of our efforts in the rebuttal.

---

### Official Review · Reviewer_dAoL · 2024-11-03

**Soundness:** 2
**Presentation:** 2
**Contribution:** 3
**Rating:** 6
**Confidence:** 3

**Summary:**

This paper proposes a novel deep learning based method to solve the missing wedge problem, which is an inverse problem that arises in cryogenic electron tomography (cryo-ET).

In cryo-ET, the goal is to reconstruct the 3D density of a biological sample (e.g. a cell) from a set of 2D projections of the density. Due to limitations during acquisition, there is a wedge-shaped region of viewing directions where no projections can be recorded. As a result, the set of projections does not uniquely determine the 3D density of the sample.

The authors propose a method called CryoGEN that aims to fill in the missing data of the missing wedge. The method is self-supervised (does not require clean ground truth for training) and fits a missing wedge reconstruction network directly to the tomogram(s) whose missing wedge is to be filled. The fitting of this reconstruction network is regularized by an energy function, represented as another neural network, which is learned together with the reconstruction network in an adversarial manner.

Based on a theoretical motivation, and experiments on synthetic data and two real-world tomograms, the authors argue that the energy function yields improved missing wedge reconstruction performance over two closely related state-of-the-art missing wedge methods (IsoNet and DeepDeWedge).


**Recommendation:** I do not recommend publishing the paper at ICLR *in its current form*. This is mainly because I found parts of the paper (especially the motivation and training of the energy function and the construction of the losses) unclear. However, since the missing wedge problem is important, and since CryoGEN performs better than state-of-the-art baselines in some cases.
I recognize the potential of the method and am willing to raise my score if the authors better explain their method and why it outperforms the baselines (IsoNet and DeepDeWedge).

**Post rebuttal:** I have increased my score from **5 to 6**, as the authors addressed most of my concerns (see discussion below). Overall, I think that the most important contribution of the paper is the observation that equipping methods that work like IsoNet or DeepDeWedge with GAN-like energy nets to can improve the missing wedge reconstruction performance.

**Strengths:**

- CryoGEN outperforms IsoNet on the clean synthetic data (Section 5.2) and gives a cleaner reconstruction of the (real) HIV capsids (Figure 10).
- The missing wedge problem addressed in the paper is important and challenging.
- The authors applied CryoGEN to two real-world tomograms, demonstrating the practical applicability of the method.

**Weaknesses:**

- I cannot really follow the motivation in Section 3, where the authors argue that IsoNet produces an average of two 3D densities $x_1$ and $x_2$ that produce the same measurement $y_0$ under the forward map $\mathcal{T}_M$. The following is unclear to me:
	- Why do the authors "choose" a Gaussian mixture prior for IsoNet (lines 170 - 171)? If I understand correctly, IsoNet does not use a prior for regularization.
	- What is the concrete energy function ($E(x) = - \log p(x)$) that gives a better solution?
	- Why does adding noise to the measurements $y$ solve the problem that there are many $x$ that map to the same $y$ (lines 194 - 199)?

- I do not understand the motivation and intuition behind the consistency loss (equation (7)). Why is the reconstruction mesh $g_\theta$ applied to y and then again after $\mathcal{T}_M$?

- The authors give no intuition as to why the energy function (which looks like a prior to me) can be learned directly on the data one wants to reconstruct. I would also appreciate more explanation as to why the energy function is learned in an adversarial manner (equation (9)).

- The authors argue that CryoGEN outperforms IsoNet on the purified ribosome tomogram (Figure 9), but I find the CryoGEN tomogram visually much less appealing. It seems to contain less detail and the contrast of the ribosomes looks too strong to me. (Since there is no ground truth, it is impossible to measure which reconstruction is actually better).

- The experiments on synthetic data (Section 5.2) are done on noiseless data. However, the low signal-to-noise ratio (as low as <10%) is one of the main challenges in cryo-ET. Therefore, it would be good to include an experiment that demonstrates how CryoGEN handles strong noise. Synthetic data is suitable for such an experiment because clean ground truth is available.

- In the abstract, the authors state that IsoNet suffers from model collapse as it updates its own training data. This aspect is not discussed in the paper.

**Questions:**

- Why is the energy function learned in an adverserial way (Equation (9))?
- Does the adverserial training of the energy function lead to any instabilities during training?
- Why does the "Posterior" loss (Equation 8) involve random rotations?

---

> ### Author Response · Authors · 2024-11-14
> **To Reviewer dAoL - Part 1**
>
> We thank the reviewer for the detailed comments and instructive feedback. We also appreciate the recognition of our contribution. We will address your remarks one by one and hope to clarify your concerns. We apologize for any lack of clarity in our current version.
>
> 1. **Motivation Clarification**:
>
>     * Gaussian Mixture Prior
>
>         * The key observation here is that for each observation $y$ (incomplete data with missing wedge), there may be multiple solutions that align the observed outcome. By "align", we mean that for a given $y$, $X|y = \arg\max p(x|y)$ can represent a set of solutions rather than a single one. In such cases, the log posterior distribution $\log p(x|y)$ can exhibit multiple maxima—a fairly common scenario.
>
>         * Our goal is to determine a parameterized function $g_\theta$ (usually a neural network) that maximizes $\log p(g_\theta(y)|y)$, which is the primary objective of this paper. In contrast, IsoNet’s objective is to directly minimize $\sum_{x_i \in X|y} ||g_\theta(y) - x_i||^2_2$, where each $x_i$ is a maximum of $\log p(x|y)$.
>
>         * The distribution $p(x|y)$ is not necessarily a Gaussian mixture; rather, it is generally assumed to be non-convex. Since the exact form of $\log p(x|y)$ is unknown, we adopt the Gaussian mixture prior as a general assumption—an approach that has proven effective in current generative tasks within deep learning (For example, diffusion models approximate the data distribution as a Gaussian mixture by convolving the discrete data with Gaussian distributions at different levels of variance, which has demonstrated strong performance).
>
>     * Lack of Ground Truth for $\log p_x(x)$
>
>         In CryoET reconstruction tasks, there is no ground truth for $\log p_x(x)$, so we approach this as an *unsupervised* learning task, unlike the concurrent work, where $p(x)$ is assumed to be known. However, we propose a way to express $p_x(x)$ by first approximating the distribution $p_y$ based on observations of the incomplete data $\{y_i\}$ and then defining $\log p_x(x) \propto \frac{1}{|\mathcal{R}|} \sum_{R \in \mathcal{R}} \log p_y(\mathcal{T}_M \circ R \circ x)$. This formulation leverages the isotropy and rotational symmetry properties of CryoET, representing a key contribution of our work.
>
>     * Addressing One-to-Many Mapping
>
>         As mentioned earlier, if the cardinality of $X|y = \arg\min p(x|y)$ is greater than one, it is impossible to find a parameterized function $g_\theta$ that can map a single observation to all possible values of $x$, as this creates a one-to-many mapping. To address this, we use $y + \epsilon$, where $\epsilon$ has full support over the real domain, giving it infinite cardinality and effectively converting the problem into a many-to-fewer mapping.
>
> 2. **Intuition Behind Consistency Loss**
>
>     This is the original objective function used in IsoNet. Although we lack reference or ground truth for the missing wedge, we do have it for other parts, enabling us to proceed with a pseudo-supervised approach. The steps of IsoNet can be outlined as follows:
>
>     * Step 1: First, we fill in the missing wedge for the incomplete data using the prediction model $g_\theta$ to generate $\tilde{x}$.
>
>     * Step 2: Next, we rotate the sample and apply a different missing wedge via $\mathcal{T}_M$ to produce $\tilde{y}$.
>
>     * Step 3: Since we have the label for the artificially created missing wedge, we can train $g_\theta$ in a supervised manner by minimizing $||\tilde{x} - g_\theta(\tilde{y})||_2^2$.
>
>     At the start of training, we incorporate this objective into our algorithm as well. The reasoning is that, when sampling from a distribution $p(x)$ using MCMC, a reasonable starting point for the Markov chain is $\int_x x p(x) \, dx$, even though this is not a minimum (see [1] for example). In our case, rather than sampling, we directly learn the mapping from $p_y$ to $p_x$ via $g_\theta$, but the intuition remains similar. Nonetheless, including this objective function still aids in improving the convergence of training $g_\theta$.

---

> ### Author Response · Authors · 2024-11-14
> **To Reviewer dAoL - Part 2**
>
> 3. **Constructing The Distribution $p_x$**
>
>     One of the key challenges—and a main contribution of our work—is constructing the distribution $p_x$ from a distribution $p_y$. We outline this process step-by-step:
>
>     * 1. Defining the Energy Function $E(y)$: Our objective is to define an energy function such that points near any data point in $\mathcal{Y} = \{y_i\}$ have low energy, while points distant from all data points have high energy. For this, contrastive learning presents itself as an effective choice. This method assigns lower $E$ values to regions with high data density and higher $E$ values elsewhere, which naturally leads us to use GANs as a tool (refer to [2] for more details).
>
>
>     * 2. Constructing $p_y$: The next question is how to construct $p_y$ given a trained energy model $E$. We address this by representing the distribution using energy-based models as $p_y(y) \propto \exp(-E(y))$, in line with a Boltzmann distribution.
>
>     * 3. Constructing $p_x$: Leveraging the isotropy property of CryoET, we define $\log p_x(x) \propto \frac{1}{|\mathcal{R}|} \sum_{R \in \mathcal{R}} \log p_y(\mathcal{T}_M \circ R \circ x)$, where $R$ denotes the rotation operation.
>
>     By integrating all these steps, we arrive at the final algorithm.
>
> 4. **Streak-like Artifacts**
>
>    For the experimental results presented in Figure 9, there is indeed no ground truth for comparison. However, we argue that the original results shown in IsoNet exhibit strong streaking artifacts. Streaking in imaging and signal processing refers to unwanted lines or artifacts in images, often caused by incomplete data. In CryoET, these artifacts arise due to the missing wedge problem, where limited angles are used for capturing images, resulting in elongated, streak-like artifacts in the reconstruction. In contrast, our method does not exhibit these streaking artifacts at all compared to IsoNet.
>
> 5. **IsoNet’s Instability**
>
>    We apologize for not explicitly addressing IsoNet’s instability. This is a general observation we have made, as IsoNet requires careful fine-tuning at each iteration. An intuitive explanation is that IsoNet’s recursive training approach can lead to error accumulation, which in turn causes instability. Specifically, in IsoNet, training $g_{\theta_t}$ depends on the predictions generated by $g_{\theta_{t-1}}$ (see our previous response - *Intuition Behind Consistency Loss*). As a result, any intermediate failure in model $g_\theta$ can propagate through subsequent cycles, ultimately compromising the entire training process. This is similar to an open-loop control system, where errors accumulate due to the lack of feedback. In contrast, our algorithm does not rely on recursive updates, resulting in faster training and greater stability.
>
> 6. **Addressing Mode Collapse in GANs**
>
>     The original GANs are known to encounter both *mode collapse* and *training instability* issues, but we address these as follows:
>
>     * Mode Collapse:
>     Mode collapse is less problematic for reconstruction in CryoET, as our goal is to find a single solution or a set of solutions $\tilde{x}$ that maximize $p(x|y)$ given the observation. While mode collapse may reduce diversity, it does not compromise quality.
>
>     * Training Instability:
>     By introducing noise $\epsilon$, our method aligns with the theoretical framework in [3] (the foundational paper on Wasserstein GANs), effectively addressing our specific one-to-many mapping problem illustrated in motivation section while significantly improving training stability.
>
> 7. **Incorporation of Rotations**
>
>    The incorporation of rotations is used to derive $p_x$ from $p_y$ by leveraging the isotropy property of CryoET.
>
> 8. **Experimental Results**
>
>    We evaluated our model and IsoNet on noisy synthetic data of Vipp1 assembly, showing that our method achieves better reconstruction of the missing wedge with reduced noise, as illustrated in Figure 16. Additionally, Figure 20 presents the visualization of a single ribosome. We also demonstrate the enhanced denoising capabilities of CryoGEN, while the comparison reveals that IsoNet consistently exhibits residual effects of the missing wedge problem across all experiments.
>
> ---
>
> **References:**
>
> [1] Chen, Wenlin, et al. "Diffusive Gibbs Sampling." *International Conference on Machine Learning*, 2024.
>
> [2] Murphy, Kevin P. "Probabilistic machine learning: Advanced topics - Chapters 24 and 26." MIT press, 2023.
>
> [3] Arjovsky, Martin, and Léon Bottou. "Towards principled methods for training generative adversarial networks." *International Conference on Learning Representations*, 2017.

---

> ### Comment · Reviewer_dAoL · 2024-11-24
> **Follow up**
>
> I thank the authors for their detailed answers to my questions. I would like to sincerely apologise for my late reply.
> Here are some comments on their answers:
>
> - **Point 4:**
> ```
> For the experimental results presented in Figure 9, there is indeed no ground truth for comparison. However, we argue that the original results shown in IsoNet exhibit strong streaking artifacts. Streaking in imaging and signal processing refers to unwanted lines or artifacts in images, often caused by incomplete data. In CryoET, these artifacts arise due to the missing wedge problem, where limited angles are used for capturing images, resulting in elongated, streak-like artifacts in the reconstruction. In contrast, our method does not exhibit these streaking artifacts at all compared to IsoNet.
> ```
>
> I did not notice streaking artefacts in the IsoNet or DeepDeWedge reconstructions. I encourage the authors to provide pointers to streaking and/or ringing (as claimed in line 444) artefacts in the IsoNet and DeepDeWedge reconstructions shown in Figure 9 and Figure 19 (a).
>
> In my opinion, the reconstruction obtained with the CryoGEN method does not look better than the IsoNet or DeepDeWedge tomograms. It just looks different. This is of course subjective because there is no ground truth.
>
> - **Point 5:**
> ```
> We apologize for not explicitly addressing IsoNet’s instability. This is a general observation we have made, as IsoNet requires careful fine-tuning at each iteration. An intuitive explanation is that IsoNet’s recursive training approach can lead to error accumulation, which in turn causes instability. Specifically, in IsoNet, training $g_{\theta_t}$ depends on the predictions generated by $g_{\theta_{t-1}}$ (see our previous response - Intuition Behind Consistency Loss). As a result, any intermediate failure in model $g_\theta$ can propagate through subsequent cycles, ultimately compromising the entire training process. This is similar to an open-loop control system, where errors accumulate due to the lack of feedback. In contrast, our algorithm does not rely on recursive updates, resulting in faster training and greater stability.
> ```
> I appreciate the authors' explanation. However, as the paper contains neither this discussion, nor an experiment that demonstrates IsoNet's claimed instability, I feel that the statement that "IsoNet relies on recursively updating its predictions, which can result in train-
> ing instability and potential model collapse." should be removed from the abstract.
>
> - **Point 7:**
> ```
> The incorporation of rotations is used to derive p_x from p_y  by leveraging the isotropy property of CryoET.
> ```
>
> I could not find any discussion of this "isotropy property" in the paper. As the authors rely on the "isotropy property" to motivate the random rotations during training, it should be discussed explicitly in the paper (e.g. in Section 4.1.1 and/or Section 4.2).

---

> > ### Author Response · Authors · 2024-11-26
> > **Follow Up to Reviewer dAoL**
> >
> > It's not late at all, and we sincerely thank you for your timely response. We greatly appreciate your detailed review of our paper and the many insightful and constructive comments you provided. We have made corresponding revisions to our manuscript and hope you find it more satisfactory.
> >
> > * **Point 4**:
> >
> >   As there is no ground truth for comparison, the evaluation is indeed subjective. To address this, we have provided additional results, allowing readers to form their own opinions more easily.
> >
> >     * Although no ground truth is available, prior research in the field has identified several common issues, particularly various types of artifacts encountered during image reconstruction. To address this, we have added a new section in the appendix, *A.4 Artifacts*, summarizing these potential artifacts and demonstrating how CryoGEN outperforms other methods.
> >
> >     * Furthermore, we agree that the original *Figure 9* is unclear. To improve clarity, we present *Figure 18*, which showcases a different slice of the ribosome and highlights streak artifacts with a zoomed-in view. Additionally, we have included a visualization of a single ribosome in *Figure 25*. The results demonstrate that CryoGEN produces smoother reconstructed outputs.
> >
> > * **Point 5**:
> >
> >   Due to time constraints, we are currently able to provide only an intuitive explanation in the revised version, specifically from *line 445* to *line 449*. However, we assure you that the final version will include a more comprehensive discussion—either empirical or theoretical—on the instability of IsoNet.
> >
> >
> > * **Point 7**:
> >
> >   We apologize for not offering a more detailed explanation of the isotropy property of CryoET. Although we have briefly mentioned it as a fundamental assumption for CryoET reconstruction in *lines 256* to *line 258*, it is not a rigorously defined characteristic. Moreover, our method has the potential to generalize to broader contexts. In the final version, we will provide a more comprehensive and rigorous mathematical motivation, introducing broader assumptions (e.g., random masking instead of random rotation), with the isotropy property included as a specific case within these general assumptions.
> >
> > Also, beyond the empirical results presented, we believe that the mathematical motivation and algorithmic framework we propose represent a significant contribution to the field of CryoET. Based on comparisons with existing published works, we are the first to address this problem comprehensively, providing a complete analysis from theoretical motivation to empirical results. Please let us know if there are any additional questions or concerns we should address. Once again, we sincerely thank you for your invaluable feedback, which has been essential in helping us enhance the quality of our work.

---

> > > ### Comment · Reviewer_dAoL · 2024-11-29
> > >
> > > Thank you for the additional clarifications. I greatly appreciate the new appendix on Artifacts!
> > >
> > > I have updated my score (see "Post rebuttal" edit in the review).

---

> > > > ### Author Response · Authors · 2024-12-03
> > > >
> > > > Thank you for taking the time to review our response. We truly value your comments, suggestions, and acknowledgment of our rebuttal.

---

### Author Response · Authors · 2024-11-21
**To Reviewers and ACs**

Dear Reviewers and ACs,

We sincerely appreciate your valuable time and effort in providing constructive feedback. We have tried our best to address your concerns and revised the paper accordingly, with additional experimental results highlighted in red. We apologize for any lack of clarity in our presentation. We are confident in the quality of our work and hope that our rebuttal has made the paper clearer and more enjoyable to read :) With a week remaining, we welcome any further constructive feedback you may have.

Thank you again.

---

### Author Response · Authors · 2024-11-28
**Follow Up**

Dear Reviewers and ACs,

We have made every effort to address all comments early in the discussion period. We understand that the volume of submissions has significantly increased this year. Nonetheless, we are confident in the quality and significance of our work and believe it merits a more in-depth discussion to clarify certain details. As the extended discussion period is ending soon, we respectfully request additional feedback for a more comprehensive evaluation of our submission.

Thanks,

---

### Meta-Review · Area_Chair_rbHL · 2024-12-23

**Metareview:**

The paper describes CryoGEN, a new reconstruction method for cryo-electron tomographic data which solves the missing-wedge problem from the acquisition of limited tilt angles in such datasets more efficiently than previous iterative methods, leading to faster and more consistent reconstructions of protein structure.

The reviewers unanimously recommended acceptance, citing the importance of the problem addressed, the novelty of the approach for CryoET, and the quality of the evaluations. Their main critiques were that:
1. similar methods have been applied to other problems in the broader field of computer vision
2. concerns about the presentation focused on insufficiently detailed explanation of the method, comparisons to previous methods, and analysis of evaluations.

These critiques were sufficiently addressed in revision.

**Additional Comments On Reviewer Discussion:**

There was sufficient back and forth during the discussion phase to enable the authors to address reviewer concerns and confusion, leading the reviewers to increase their scores to acceptance.

---

### Decision · Program_Chairs · 2025-01-22

Accept (Poster)